# MODEL PREDICTIVE CONTROL IS ALMOST OPTIMAL FOR RESTLESS BANDITS

## ABSTRACT

We consider the discrete time infinite horizon average reward restless markovian bandit (RMAB) problem. We propose a *model predictive control* based non-stationary policy with a rolling computational horizon $\tau$. At each time-slot, this policy solves a $\tau$ horizon linear program whose first control value is kept as a control for the RMAB. Our solution requires minimal assumptions and quantifies the loss in optimality in terms of $\tau$ and the number of arms, $N$. We show that its sub-optimality gap is $O(1/\sqrt{N})$ in general, and $\exp(-\Omega(N))$ under a local-stability condition. Our proof is based on a framework from dynamic control known as *dissipativity*. Our solution is easy to implement and performs very well in practice when compared to the state of the art. Further, both our solution and our proof methodology can easily be generalized to more general constrained MDP settings and should thus, be of great interest to the burgeoning RMAB community.

## 1 INTRODUCTION

This work investigates the sequential decision making problem of Restless Multi-Armed Bandits (RMAB for short) over an infinite discrete time horizon. In this problem there are $N$ statistically identical arms. At each time step, the decision maker must choose for each arm if they would like to pull the arm or leave it as is. The decision maker has a constraint $\alpha N$ on the maximal number of arms that they may pull at each time instance. Each arm has a known state belonging to a common finite state space and upon choosing an action, produces a known state and action dependent reward. Next, the arms evolve to a new state independently according to a known state-action dependent transition kernel. These arms are only coupled through the budget constraint on the number of arms that maybe pulled at each time instance. The state and reward are both revealed to the decision maker before the next decision needs to be made. The objective of the decision maker is to maximize the long term time average reward.

This problem was first proposed by Whittle (1988). Over the years RMABs have been used to model a number of practical problems. These applications include web-crawling, queuing, communication systems, scheduling problems and many more, (Veatch & Wein, 1996), (Dance & Silander, 2019), (Nino-Mora, 2002), (O'Meara & Patel, 2001). The problem of choosing a subset of tasks to perform among a larger collection of tasks under resource constraints shows up time and time again in various resource constrained control problems. For a comprehensive review on RMABs and their applications, the interested reader is directed towards Niño-Mora (2023). While the existence of an optimal policy for RMABs straightforward, Papadimitriou & Tsitsiklis (1999) showed that the exact solution to this problem is PSPACE-hard. Consequently, most work focused on designing approximate solutions with good performance guarantees.

In the seminal paper, Whittle (1988) suggested that under a condition known as indexability, an index can be associated with each state. This index is now referred to as the Whittle's index (WI) and it was conjectured that a priority policy based on this index would be an optimal solution for this problem. This setting naturally lends itself to mean field approximations where one may replace the $N$ armed problem with a dynamical system in order to find these approximate solutions. (Weber & Weiss, 1990) were the first to point out that under a (hard to verify) condition on the dynamical system known as uniform global attractor property (UGAP), the (WI) was asymptotically optimal. Recently, many of the results in the RMAB literature have focused on two major aspects of the problem. Firstly, how quickly do proposed asymptotically optimal policies converge to the optimal solution

as a function of the number of arms. Secondly, can the underlying assumptions such as indexability and UGAP be made less restrictive to obtain more general conditions under which optimal solutions can be found. In the former category, under the assumptions of Weber & Weiss (1990), Gast et al. (2023a) showed that the WI policy is exponentially close to the optimal solution. More recently still, several works have been able to show an exponential order of convergence, Gast et al. (2023b); Hong et al. (2024a) under less restrictive assumptions. On the other hand, in the latter category, Hong et al. (2023) showed an asymptotic convergence result under a far less restrictive assumption known as the *synchronization assumption*. Similar works following this steering dynamics include Yan (2024) and Hong et al. (2024b) to show asymptotically optimal algorithms. Furthermore, Hong et al. (2024a) loosened the restrictions further to not only show asymptotic convergence but describe fundamental conditions in order to achieve exponential convergence rates for any algorithm. A more comprehensive view of recent results can be found in Appendix A with descriptions of both assumptions and the algorithm.

In this work, we return to the dynamical control perspective but do not look to steer our system towards an optimal fixed point. Instead, we focus on the notion of *dissipativity*, to make the case for a well known set of policies known as *model predictive control*. We show that a simple MPC with a finite planning horizon produces an asymptotically optimal solution to the RMAB problem without any need to relax constraints. In doing so, not only do we bring a new perspective to this decades old problem but also describe a simple tractable algorithm that performs exceptionally well in practice with a small planning horizon.

**Main Contributions**   The main contribution of our paper is the proof that a very natural model predictive control (LP-update) provides both an algorithm that works well in practice as well as theoretical guarantees *under very weak and easy to verify assumptions on the system parameters only,* using the framework of dissipativity. The idea of resolving an LP for finite-horizon restless bandits already exists in the literature, however, all the papers that proposed to use this idea either analyzed the algorithm in the finite-horizon case or used the hard to verify UGAP assumption(*uniform global attractor property*) on the system. This is primarily because without the framework of dissipativity, analyzing how a finite horizon policy translates to the infinite horizon reward seems to be infeasible. The use of this framework is one of the key technical novelties of our approach. A greater exposition on our work's place in the current literature is relegated to the Related works section, Appendix A due to lack of space. Apart from the main result, we make several fresh observations about the nature of RMAB problems:

1. Our proof is of independent interest since it uses a new framework known as *dissipativity*. Dissipativity is a closely studied phenomenon in the model predictive literature and is used to study how a policy drives a system towards optimal fixed points Damm et al. (2014).

2. Returning to the dynamics around the fixed point, we can tighten the rate of convergence to $e^{-cN}$ under a local stability condition.

3. Perhaps the most helpful portion of our results, for practical purposes; the MPC algorithm works well in practice and is easy to implement. It performs well both in terms of the number of arms $N$ as well as the computational time horizon $T$ beating state of the art algorithms in our bench marks.

**Road-map**   The rest of the paper is organized as follows. We describe the system model and the corresponding linear relaxation in Section 2. We build the LP-update algorithm in Section 3 and present its performance guarantee in Section 4. We provide the main ingredients of the proof in Section 5 postponing the most technical lemmas to the appendix. We illustrate the performance of the algorithm in Section 6. The appendix contains additional literature review A, details of the algorithm and their extension to multi-constraints MDPs B, additional proofs C, D and details about the numerical experiments F.

## 2   SYSTEM MODEL AND LINEAR RELAXATION

### 2.1   SYSTEM MODEL

We consider an infinite horizon discrete time restless Markovian bandit problem parameterized by the tuple $\langle \mathcal{S}, \mathbf{P}^0, \mathbf{P}^1, \mathbf{r}^0, \mathbf{r}^1; \alpha, N \rangle$. A decision maker is facing $N$ statistically identical arms, and

each arm has a state that belongs to the finite state-space $\mathcal{S}$. At each time-instant, the decision maker observes the states of all arms $\mathbf{s} = \{s_1 \ldots s_N\}$ and chooses a vector of actions $\mathbf{a} = \{a_1 \ldots a_N\}$, where the action $a_n = 1$ (respectively $a_n = 0$) corresponds to pulling the arm $n$ (or to leave it). The decision maker is constrained to pull at most $\alpha N$ arms at each decision epoch.

The matrices $\mathbf{P}^0$ and $\mathbf{P}^1$ denote the transitions matrices of each arm and the vectors $\mathbf{r}^0, \mathbf{r}^1$ denote the $|\mathcal{S}|$ dimensional vector for the rewards. We assume that all the rewards $\mathbf{r}$ lie between $0$ and $1$. As the state-space is finite, this assumption can be made without loss of generality by scaling and centering the reward vector. We assume that the transitions of all arms are Markovian and independent. This means that if the arms are in state $\mathbf{s} \in \mathcal{S}^N$ and the decision maker takes an action $\mathbf{a} \in \{0,1\}^N$, then the decision maker earns a reward $\sum_n r_{a_n}^n$ and the next state becomes $\mathbf{s}' \in \mathcal{S}^N$ with probability

$$\mathbb{P}(\mathbf{S}(t+1) = \mathbf{s}'|\mathbf{S}(t) = \mathbf{s}, \mathbf{A}(t) = \mathbf{a}, \ldots \mathbf{S}(0), \mathbf{A}(0))$$
$$= \mathbb{P}(\mathbf{S}(t+1) = \mathbf{s}'|\mathbf{S}(t) = \mathbf{s}, \mathbf{A}(t) = \mathbf{a}) = \Pi_{n=1}^N P_{s_n, s'_n}^{a_n}. \tag{1}$$

It is important to note that the arms are only coupled through the budget constraint $\alpha$.

Let $\pi$ be a stationary policy mapping each state to a probability of choosing actions, i.e, $\pi : \mathcal{S}^N \to \Delta(\{0,1\}^N)$ subject to the budget constraint $(\alpha)$. Let $\Pi^{(N)}$ denote the space of all such policies. Given a policy $\pi$, mapping the joint state of the arms to a realization of the $N$ length joint action vector at time $t$, $\mathbf{A}^N(t) := \{a_1(t), a_2(t) \ldots a_N(t)\}$, and an initial set of arm states $\mathbf{S}^{(N)}(0) = \mathbf{s}$, we define the average gain of policy $\pi$ as

$$V_\pi^{(N)}(\mathbf{s}) = \lim_{T \to \infty} \frac{1}{T} \mathbb{E}_\pi \left[ \sum_{t=0}^{T-1} \frac{1}{N} \sum_{n=1}^N r_{S_n^N(t)}^{A_n(t)} \middle| \mathbf{S}^{(N)}(0) = \mathbf{s} \right]. \tag{2}$$

Here $(S_n^N(t), A_n^N(t))$ denotes the state-action pair of the $n$th arm at time $t$ and $r_{S_n^N(t)}^{A_n^N(t)}$ denotes the $S_n^N(t)^{\text{th}}$ entry of the $\mathbf{r}^{A_n(t)}$ vector. As the state-space and action space is finite, for any stationary policy $\pi$, the limit is well defined Puterman (2014). The RMAB problem amounts to computing a policy $\pi$ that maximizes the infinite average reward. We denote the optimal value of the problem as

$$V_{\text{opt}}^N(\mathbf{s}) := \max_{\pi \in \Pi^{(N)}} V_\pi^{(N)}(\mathbf{s}). \tag{3}$$

The optimal policy exists and the limit is well defined, Puterman (2014). Under mild conditions (which will be verified in our case), this value does not depend on the initial state in which case we will simply write it as $V_{\text{opt}}^N$. Note, by restricting our policies to $\Pi^{(N)}$ we are enforcing the budget constraints at each time step.

## 2.2 ALTERNATIVE STATE REPRESENTATION VIA EMPIRICAL DISTRIBUTION

In order to build an approximation of (3), we introduce an alternative representation of the state space, that we will use extensively in the paper. Given any joint state of the arms $\mathbf{s} \in \mathcal{S}^N$, we denotes the empirical distribution of these arms as $\mathbf{x}(\mathbf{s}) \in \Delta_S$, where $\Delta_{\mathcal{S}}$ is the simplex of dimension $|\mathcal{S}|$. $\mathbf{x}(\mathbf{s})$ is a vector with $|\mathcal{S}|$ dimensions and $x_i(\mathbf{s})$ is the fraction of arms that are in state $i$. Next, given an action vector $\mathbf{a}$, we denote by $\boldsymbol{u}(\mathbf{s}, \mathbf{a})$ the empirical distribution of the state-action pairs $(s, 1)$. In words, $u_i(\mathbf{s}, \mathbf{a})$ is the fraction of arms that are in state $i$ and that are pulled. Since no more than $N\alpha$ arms can be pulled at any time instance and no more than $N\mathbf{x}_s$ ($\mathbf{x}_s$ is the fraction of arms in state $s$) arms can be pulled in state $s$, it follows that when $\mathbf{x}$ is fixed, $\boldsymbol{u}$ satisfies the following inequalities,

$$0 \le \boldsymbol{u} \le \mathbf{x} \qquad \|\boldsymbol{u}\|_1 \le \alpha \|\mathbf{x}\|_1 = \alpha, \tag{4}$$

where $\boldsymbol{u} \le \mathbf{x}$ denotes a component-wise inequality and $\| \cdot \|_1$ denotes the $l_1$ norm. We denote by $\mathcal{U}(\mathbf{x})$ the set of feasible actions for a given $\mathbf{x}$, *i.e.*, the set of $\boldsymbol{u}$ that satisfy (4).

## 2.3 LINEAR RELAXATION

We consider the following linear program:

$$V_{\text{opt}}^\infty := \max_{\mathbf{x}, \boldsymbol{u} \in \Delta_{\mathcal{S}}} \mathbf{r}^0 \cdot \mathbf{x} + (\mathbf{r}^1 - \mathbf{r}^0) \cdot \boldsymbol{u}, \tag{5a}$$

$$\text{Subject to:} \quad \boldsymbol{u} \in \mathcal{U}(\mathbf{x}) \tag{5b}$$

$$\mathbf{x} = \mathbf{x}\mathbf{P}^0 + \boldsymbol{u}(\mathbf{P}^1 - \mathbf{P}^0) \tag{5c}$$

This linear program is known to be a relaxation of (3) that is asymptotically tight, that is $V_{\text{opt}}^N \leq V_{\text{opt}}^\infty$ for all $N$ and $\lim_{N\to\infty} V_{\text{opt}}^N = V_{\text{opt}}^\infty$, see Gast et al. (2023b); Hong et al. (2024a).

To give some intuition on the relationship between (3) and (5), we remark that if $\boldsymbol{X}(t) := \mathbf{x}(\mathbf{S}^N(t))$ is the empirical distribution of states at time $t$ and $\boldsymbol{U}(t) = \boldsymbol{u}(\mathbf{S}(t), \mathbf{A}(t))$ is the joint control, then it is shown in Gast et al. (2023b) that the Markovian evolution (1) implies

$$\mathbb{E}[\boldsymbol{X}(t+1) \mid \boldsymbol{X}(t), \boldsymbol{U}(t)] = \boldsymbol{X}(t)\mathbf{P}^0 + \boldsymbol{U}(t)(\mathbf{P}^1 - \mathbf{P}^0). \tag{6}$$

In (5), the variable $x_i$ corresponds to the time-averaged fraction of arms in state $i$ and similarly the variable $u_i$ corresponds to the time-averaged fraction of arms in state $i$ that are pulled. The constraint (5b) imposes that *on average*, no more than $\alpha N$ arms are pulled. This is in contrast with the condition imposed for problem (3) that enforces this condition at each time step.

## 3 CONSTRUCTION OF THE LP-UPDATE POLICY

### 3.1 THE FINITE-HORIZON MEAN FIELD CONTROL PROBLEM

To build the LP-update policy, we consider a *controlled dynamical system*, also called the *mean field model*, that is a finite-time equivalent of (5). For a given initial condition $\boldsymbol{x}(0)$ and a time-horizon $\tau$, the states and actions of this dynamical system are constrained by the evolution equations

$$\boldsymbol{u}(t) \in \mathcal{U}(\boldsymbol{x}(t)) \tag{7a}$$

$$\boldsymbol{x}(t+1) = \boldsymbol{x}(t)\mathbf{P}^0 + \boldsymbol{u}(t)(\mathbf{P}^1 - \mathbf{P}^0), \tag{7b}$$

$\forall t \in \{0 \ldots \tau - 1\}$. In the above equation, (7b) should be compared with (6) and indicates that $\boldsymbol{x}(t)$ and $\boldsymbol{u}(t)$ correspond to the quantities $\mathbb{E}[\boldsymbol{X}(t)]$ and $\mathbb{E}[\boldsymbol{U}(t)]$ of the original stochastic system. As the constraint (7a) must be ensured by $\boldsymbol{x}(t)$ and $\boldsymbol{u}(t)$, this constraint (4) must be satisfied for the expectations: $\mathbb{E}[\boldsymbol{U}(t)] \in \mathcal{U}(\mathbb{E}[\boldsymbol{X}(t)])$.

The reward collected at time $t$ for this dynamical system is $\mathbf{r}^0 \cdot \boldsymbol{x}(t) + (\mathbf{r}^1 - \mathbf{r}^0) \cdot \boldsymbol{u}(t)$. Let $\lambda$ be the dual multiplier of the constraint (5c) of an optimal solution of (5). We define a *deterministic* finite-horizon optimal control problem as:

$$W_\tau(\mathbf{x}(0)) = \max_{\boldsymbol{x},\boldsymbol{u}} \sum_{t=0}^{\tau-1} \left( \mathbf{r}^0 \cdot \mathbf{x}(t) + (\mathbf{r}^1 - \mathbf{r}^0) \cdot \boldsymbol{u}(t) \right) + \lambda \cdot \boldsymbol{x}(\tau), \tag{8a}$$

$$\text{Subject to: } \boldsymbol{x} \text{ and } \boldsymbol{u} \text{ satisfy (7) for all } t \in \{0, \tau - 1\}, \tag{8b}$$

Before moving forward, the above equation deserves some remarks. First, for any finite $\tau$, the objective and the constraints (7) of the optimization problem (8) are linear in the variables $(\boldsymbol{x}(t), \boldsymbol{u}(t))$. This means that this optimization problem is computationally easy to solve. In what follows, we denote by $\mu_\tau(\mathbf{x})$ the value of $\boldsymbol{u}(0)$ of an optimal solution to (8).

Second, the definition of (8) imposes that the constraint $\|\boldsymbol{u}\|_1 \leq \alpha \|\mathbf{x}\|_1 = \alpha$ holds for each time $t$. This is in contrast to the way this constraint is typically treated in RMAB problems, in which case (4) is replaced with the time-averaged constraint $\frac{1}{T} \sum_{t=0}^{T-1} \|\boldsymbol{u}(t)\|_1 \leq \alpha$. The latter relaxation was introduced in Whittle (1988) and is often referred to as Whittle's relaxation (Avrachenkov & Borkar, 2020; Avrachenkov et al., 2021). This is the constraint that we use to write (5). Gast et al. (2023a) showed that for any finite $T$, the finite-horizon equivalent of (3) converges to (8) as $N$ goes to infinity. The purpose of this paper is to show that the solution of the finite $T-$horizon LP (8) provides an almost-optimal solution to the original $N-$arm average reward problem (3).

Last, as we will discuss later, taking $\lambda$ as the dual multiplier of the constraint (5c) helps to make a connection between the finite and the infinite-horizon problems (8) and (5). Our proofs will hold with minor modification by replacing $\lambda$ by 0 and in practice we do not use this multiplier for our algorithm.

### 3.2 THE MODEL PREDICTIVE CONTROL ALGORITHM

The pseudo-code of the LP-update policy is presented in Algorithm 1. The LP-update policy takes as an input a time-horizon $\tau$. At each time-slot, the policy solves the finite horizon linear program

---

**Algorithm 1** Evaluation of the LP-Update policy

---

**Input:** Horizon $\tau$, Initial state $\mathbf{S}^{(N)}(0)$, model parameters $\langle \mathbf{P}^0, \mathbf{P}^1, \mathbf{r}^0, \mathbf{r}^1 \rangle$, and time horizon $T$

    Total-reward $\leftarrow 0$.
    **for** $t = 0$ to $T - 1$ **do**
        $\boldsymbol{u}(t) \leftarrow \mu_\tau(\mathbf{x}(\mathbf{S}^{(N)}(t)))$.
        $\mathbf{A}^{(N)}(t) \leftarrow$ Randomized Rounding $(\boldsymbol{u}(t))$ (by using Algorithm 2).
        Total-reward $\leftarrow$ Total-reward $+ R(\mathbf{S}^{(N)}(t), \mathbf{A}^{(N)}(t))$.
        Simulate the transitions according to (1) to get $\mathbf{S}^{(N)}(t+1)$
    **end for**
**Output:** Average reward : $\frac{\text{Total-reward}}{T} \approx V_{\text{LP},\tau}^{(N)}(\infty)$

---

(8) to obtain $\mu_\tau(\mathbf{x})$ that is the value of $\boldsymbol{u}(0)$ of an optimal solution to (8). Note that such a policy may not immediately translate to an applicable policy as we do not require that $N\mu_\tau(\mathbf{x})$ to be integers. We therefore use *randomized rounding* to obtain a feasible policy for our $N$ armed problem, $\mathbf{A}^N(t)$. Applying these actions to each arm gives an instantaneous reward and a next state. This form of control has been referred to as *rolling horizon* Puterman (2014) but more commonly referred to as *model predictive control* Damm et al. (2014). Our algorithm maybe summarized more succinctly as:

$$\mathbf{S}^{(N)}(t) \xrightarrow{\text{Solve LP (8)}} \mu_\tau(\mathbf{x}(\mathbf{S}^{(N)}(t))) \xrightarrow[\text{rounding}]{\text{Randomized}} \mathbf{A}^N(t) \xrightarrow[\text{new state}]{\text{Observe}} \mathbf{S}^{(N)}(t+1). \tag{9}$$

The randomized rounding procedure that we use is similar to the one described in Gast et al. (2023a). We discuss it in Appendix B.1.

## 4 MAIN THEORETICAL RESULTS

The main result of our paper is to show that a finite-horizon model predictive control algorithm, that we call the LP-update policy, is asymptotically optimal for the infinite horizon bandit problem. Note that this LP-update policy is introduced in Gast et al. (2023a;b); Ghosh et al. (2022) for finite-horizon restless bandit.

### 4.1 FIRST RESULT: LP-UPDATE IS ASYMPTOTIC OPTIMAL

Our result will show that the LP-update policy is asymptotically optimal as the number of arms $N$ goes to infinity, under an easily verifiable mixing assumption on the transition matrices. To express this condition, for a fixed integer and a sequence of action $\mathbf{a} = (a_1 \ldots a_k) \in \{0,1\}^k$, we denote by $P_{i,j}^{\mathbf{a}}$ the $(i,j)$th entry of the matrix $\prod_{t=1}^k \mathbf{P}^{a_k}$. We then denote by $\rho_k$ the following quantity[1]:

$$\rho_k \triangleq \min_{s,s' \in \mathcal{S}, \mathbf{a} \in \{0,1\}^k} \sum_{s^* \in \mathcal{S}} \min\{P_{s,s^*}^{a_1,a_2,\ldots a_k}, P_{s',s^*}^{0,0,\ldots 0}\} \tag{10}$$

In the above equation, the minimum is taken over all possible initial states $s, s'$ and all possible sequence of actions. The quantity $\rho_k$ can be viewed as the probability (under the best coupling) that two arms starting in states $s$ and $s'$ reach the same state after $k$ iterations, if the sequence $a_1 \ldots a_k$ is used for the first arm while the second arm only uses the action $0$. The assumption that we use for our result is that $\rho_k > 0$ for some integer $k$.

**Assumption 1.** *There exists a finite $k$ such that $\rho_k > 0$.*

While the assumption may look abstract, note that when the $\mathbf{P}^0$ matrix is ergodic, it ensures that assumption 1 holds. Indeed in this case, there exists a $k > 0$ such that $P_{ij}^{0 \ldots 0} > 0$ for all $i, j$ which would imply that $\rho_k > 0$. Related assumptions and their relationship to Ergodicity can be found in Hernandez-Lerma & Lasserre (2012). Assumption 1 is similar to the unichain and aperiodic condition imposed in Hong et al. (2024b). Note that as this quantity involves the best coupling and

---

[1]The definition (10) is related to the notion of ergodic coefficient defined in (Puterman, 2014) that used a related constant to prove convergence of value iteration algorithms in span norm for the unconstrained discounted Markov Decision Process.

not a specific coupling, it is more general than the synchronization assumption used in Hong et al. (2023).

We are now ready to state our first theorem, where we provide a performance bound of the average reward of the LP-update policy, that we denote by $V_{\text{LP},\tau}^{(N)}(\infty)$.

**Theorem 4.1.** *Assume 1. There exist constants $C_\lambda, C_\Phi > 0$ such that for any $\epsilon > 0$, there exists $\tau(\epsilon)$ such that, Algorithm 1 has the following guarantee of performance:*

$$V_{LP,\tau(\epsilon)}^{(N)}(\infty) \geq V_{opt}^{N}(\infty) - \epsilon - \left( \frac{kC_\Phi}{\rho} + C_\lambda C_\Phi + 1 \right) \frac{(\alpha N - \lfloor \alpha N \rfloor)}{N} - \left( \frac{k}{\rho_k} + C_\lambda \right) \left( \sqrt{\frac{|\mathcal{S}|}{N}} \right) \tag{11}$$

This result shows that the LP-update policy becomes optimal as $\tau$ and $N$ go to infinity. The sub-optimality gap of LP-update decomposes in three terms. The first term corresponds to an upper-bound on the sub-optimality of using the finite-horizon $\tau$ when solving the LP-problem (8). Note, our proof shows that one can take $\tau(\epsilon) = \mathcal{O}(\frac{1}{\epsilon})$. Moreover, in the numerical section, we will show that choosing a small value like $\tau = 10$ is sufficient for most problems. The second term corresponds to a rounding error: for all $\alpha$ such that $N\alpha$ is an integer, this term equals 0. The dominating error term is the last term, $O(1/\sqrt{N})$, corresponding to the difference between the $N$-arm problem and the LP-problem with $\tau + \infty$.

## 4.2 SECOND RESULT: EXPONENTIALLY SMALL GAP UNDER A STABILITY CONDITION

Theorem 4.1 shows that the sub-optimality gap of the LP-update policy is of order $O(1/\sqrt{N})$ under quite general conditions. While one could not hope for a better convergence rate in general, there are known cases for which one can construct policies that become optimal exponentially fast when $N$ goes to infinity. This is the case for Whittle index under the conditions of indexability, uniform global attractor property (UGAP), non-degeneracy and global exponential stability Gast et al. (2023a). More details can be found in Appendix A. In this section, we show that LP-update also becomes optimal exponentially fast under essentially the same conditions as the ones presented in Hong et al. (2024a). The first condition that we impose is that the solution of the above LP-problem is non-degenerate (as defined in Gast et al. (2023a); Hong et al. (2024a)).

**Assumption 2** (Non-degenerate). *We assume that the solution $(\boldsymbol{x}^*, \boldsymbol{u}^*)$ to the linear program (5) is unique and satisfies that $x_i^* > 0$ for all $i \in \mathcal{S}$ and that there exists a (unique) state $i^* \in \mathcal{S}$ such that $0 < u_{i^*}^* < x_{i^*}^*$.*

The second condition concerns the local stability of a map around the fixed point.

**Assumption 3.** *Assume 2 and let $P^*$ be the $|\mathcal{S}| \times |\mathcal{S}|$ matrix such that*

$$P_{ij}^* = \begin{cases} P_{ij}^0 & \text{if } i \text{ is such that } u_i^* < x_i^*. \\ P_{ij}^1 - P_{i^*j}^1 + P_{i^*j}^0 & \text{if } i \text{ is such that } u_i^* = x_i^*. \end{cases}$$

*We assume that the matrix $P^*$ is stable, i.e., that the $l_2$ norm of all but one of the Eigenvalues of $\mathbf{P}^*$ are strictly smaller than 1.*

Both these conditions are equivalent to the assumption of non-degeneracy and local stability defined in Hong et al. (2024a).

The last condition that we impose is a technical assumption that simplifies the proofs.

**Assumption 4** (Unicity). *We assume that for all $\boldsymbol{x} \in \Delta_\mathcal{S}$, the LP program (8) has a unique solution.*

This assumption guarantees that the LP-update policy is uniquely defined. Note that the assumptions of unicity of the fixed point are often made implicitly in papers when authors talk about "the" optimal solution instead of "an" optimal solution.

**Theorem 4.2.** *Assume 1, 2, 3, and 4 then there exist constants $C', C'' > 0$ (independent on $N$ and $\tau$) such that for all $\epsilon > 0$, with $\tau(\epsilon)$ (set according to Theorem 4.1) and $N$ such that $\alpha N$ is an integer, Algorithm 1 has the following guarantee of performance*

$$V_{LP,\tau(\epsilon)}^{(N)}(\infty) \geq V_{opt}^{N}(\infty) - \epsilon - C'e^{-C''N}. \tag{12}$$

The first term of the bound above is identical to the one used in Theorem 4.1. What is more important is that the last term decays exponentially with $N$.

## 5 PROOFS: MAIN IDEAS

In this section, we provide the major ingredients of the proofs of the two main theorems. We provide more details for the proof of Theorem 4.1 because this is the more original of the two. The proofs of all lemmas and some details of computation are deferred to Appendix C.

### 5.1 SKETCH FOR THEOREM 4.1:

Three major components are required in-order to complete the proof.

**Part 1, Properties of the dynamical control problem** (8)   For $\boldsymbol{x}, \boldsymbol{u}$, we denote by $\Phi(\boldsymbol{x}, \boldsymbol{u}) := \boldsymbol{x}\mathbf{P}^0 + \boldsymbol{u}(\mathbf{P}^1 - \mathbf{P}^0)$ the deterministic transition kernel, and we recall that the instantaneous reward is $R(\boldsymbol{x}, \boldsymbol{u}) := \mathbf{r}^0 \cdot \boldsymbol{x} + (\mathbf{r}^1 - \mathbf{r}^0)$. In Lemma C.2, we establish several properties that relate the finite-horizon problem (8) and the finite-horizon problem (5) that hold under Assumption 1. First we show that the average gain of the finite-time horizon problem (8) converges to the average gain of the infinite-horizon problem, that is $\lim_{\tau \to \infty} W_\tau(\boldsymbol{x})/\tau = g^*$ for all $\boldsymbol{x}$. Second, we also show that the *bias function* $h^\star(\cdot) : \Delta_{\mathcal{S}} \to \mathbb{R}$ given by :

$$h^\star(\boldsymbol{x}) := \lim_{\tau \to \infty} W_\tau(\boldsymbol{x}) - \tau g^\star \qquad (13)$$

is well defined and Lipschitz-continuous with constant $k/\rho_k$ and the convergence in (13) is uniform in $\boldsymbol{x}$. Moreover, the gain and the bias function satisfy

$$h^\star(\boldsymbol{x}) + g^\star = \max_{\boldsymbol{u} \in \mathcal{U}(\boldsymbol{x})} R(\boldsymbol{x}, \boldsymbol{u}) + h^\star(\Phi(\mathbf{x}, \boldsymbol{u})). \qquad (14)$$

While both these definitions are well known in the average reward *unichain MDP* setting *without constraints*, Lemma C.2 establishes these definitions for the *constrained, deterministic problem*.

**Part 2, *Dissipativity* and *rotated cost*** Let $(\boldsymbol{x}^*, \boldsymbol{u}^*)$ be the[2] optimal solution of the infinite-horizon problem (5), and let $l(\mathbf{x}, \boldsymbol{u}) := g^\star - \mathbf{r}^0 \cdot \mathbf{x} - (\mathbf{r}^1 - \mathbf{r}^0) \cdot \boldsymbol{u}$. Following Damm et al. (2014), an optimal control problem with stage cost $l(\boldsymbol{x}, \boldsymbol{u})$ and dynamic $\boldsymbol{x}(t+1) := \Phi(\boldsymbol{x}, \boldsymbol{u})$ is called *dissipative* if there exists a *storage function* $\lambda : \Delta_{\mathcal{S}} \to \mathbb{R}$ that satisfies the following equation:

$$\tilde{l}(\boldsymbol{x}, \boldsymbol{u}) := l(\mathbf{x}, \boldsymbol{u}) + \lambda(\mathbf{x}) - \lambda(\Phi(\mathbf{x}, \boldsymbol{u})) \geq l(\boldsymbol{x}^*, \boldsymbol{u}^*) = \tilde{l}(\boldsymbol{x}^*, \boldsymbol{u}^*) = 0.$$

The cost, $\tilde{l}(\mathbf{x}, \boldsymbol{u})$ is called the *rotated cost function*.

In Lemma C.3, we show that our problem is dissipative by setting the storage function $\lambda(x) := \lambda \cdot x$, where $\lambda$ is the optimal dual multiplier of the constraint (8a). It is important to note, the rotated cost so defined is always non-negative.

**Part 3, MPC is optimal for the deterministic control problem**   By using our definition of rotated cost, we define the following minimization problem

$$L_\tau(\mathbf{x}) := \min \sum_{t=0}^{\tau-1} \tilde{l}(\mathbf{x}(t), \boldsymbol{u}(t)).$$

$$\text{Subject to: } \boldsymbol{x}(t) \text{ and } \boldsymbol{u}(t) \text{ satisfy (7) for all } t \in \{0, T-1\}. \qquad (15)$$

By dissipativity, $L_\tau(\mathbf{x})$ is monotone increasing. Moreover, we have that $L_T(\mathbf{x}) \geq 0 = L_T(\boldsymbol{x}^*)$. Hence, the problem is *operated optimally at the fixed point*, $(x^*, u^*)$. The optimal operation at a fixed point is a key observation, made by several works, (Goldsztajn & Avrachenkov, 2024; Gast et al., 2023b; Yan, 2024; Hong et al., 2023), we recover this result as a natural consequence of dissipativity. By Lemma C.4, $L_\tau(\mathbf{x}) = \tau g^\star - W_\tau(\mathbf{x}) + \lambda \cdot \mathbf{x}$ because[3] of a telescopic sum of the terms $\lambda \cdot \boldsymbol{x}(t)$. Combining this with (13) implies that $\lim_{\tau \to \infty} L_\tau(\mathbf{x}) = h^\star(\boldsymbol{x}) + \lambda \cdot \boldsymbol{x}$. As $L_\tau(\boldsymbol{x})$ is monotone, it follows that, for any $\epsilon > 0$, there exists a $\tau(\epsilon)$ such that for $L_{\tau(\epsilon)}(\boldsymbol{x}) - L_{\tau(\epsilon)-1}(\boldsymbol{x}) < \epsilon$. Note that the fact that $L_\tau(\boldsymbol{x})$ is monotone is crucial to obtain this property, leading us to use dissipativity. Putting the three components together, we can now prove the final steps.

*Proof of Theorem 4.1.* We begin by considering the difference between the optimal value and the model predictive value function. We let $\boldsymbol{X}(t)$ denote the empirical distribution of the arms at time

---

[2]For clarity we will present the proof as if this point is unique although the proof can be rendered without this requirement.

[3]In fact, the two control problems are equivalent: $\boldsymbol{u}^*$ is optimal for (8) if and only if it is for (15).

$t$ and $\boldsymbol{U}(t)$ be the corresponding empirical distribution of the actions. We drop the superscript $N$ for convenience in the proof below, but it should be noted that $\boldsymbol{U}(t)$ is always obtained from a randomized rounding procedure and hence, dependent on $N$.

$$V_{\text{opt}}^N(\mathbf{x}) - V_{\text{LP},T}^{(N)}(\mathbf{x}) \leq \lim_{T \to \infty} \frac{1}{\tau} \sum_{t=0}^{T-1} \mathbb{E}\left[g^\star - R(\boldsymbol{X}(t), \boldsymbol{U}(t))\right]$$

$$= \lim_{T \to \infty} \frac{1}{T} \sum_{t=0}^{T-1} \mathbb{E}\left[g^\star - \mathbf{r}^0 \cdot \boldsymbol{X}(t) - (\mathbf{r}^1 - \mathbf{r}^0) \cdot \boldsymbol{u}(t) + (\mathbf{r}^1 - \mathbf{r}^0) \cdot (\boldsymbol{u}(t) - \boldsymbol{U}(t))\right]$$

$$\leq \lim_{T \to \infty} \frac{1}{T} \sum_{t=0}^{T-1} \mathbb{E}\left[l(\boldsymbol{X}(t), \boldsymbol{u}(t))\right] + \frac{\alpha N - \lfloor \alpha N \rfloor}{N} \tag{16}$$

The first inequality follows from the well known result $V_{\text{opt}}^N(\mathbf{x}) \leq g^\star$, (Yan, 2024; Gast et al., 2023a; Hong et al., 2023). The last inequality follows from randomized rounding and Lemma B.1. Let $(A) := \lim_{T \to \infty} \frac{1}{\tau} \sum_{t=0}^{T-1} \mathbb{E}\left[l(\boldsymbol{X}(t), \boldsymbol{u}(t))\right]$ denotes the first term of (16). Adding and subtracting the storage cost we have:

$$(A) = \lim_{T \to \infty} \frac{1}{T} \sum_{t=0}^{T-1} \mathbb{E}\left[\tilde{l}(\boldsymbol{X}(t), \boldsymbol{u}(t)) - \lambda \cdot \boldsymbol{X}(t) + \lambda \cdot \Phi(\boldsymbol{X}(t), \boldsymbol{u}(t))\right].$$

Now note, by the dynamic programming principle $\tilde{l}(\boldsymbol{x}, \boldsymbol{u}) = L_\tau(\boldsymbol{x}) - L_{\tau-1}(\Phi(\boldsymbol{x}, \boldsymbol{u}))$. Further, for any state $\boldsymbol{x}$ and its corresponding control $\boldsymbol{u}$ consider:

$$L_\tau(\boldsymbol{x}) - L_{\tau-1}(\Phi(\boldsymbol{x}, \boldsymbol{u})) = L_\tau(\boldsymbol{x}) - L_\tau(\Phi(\boldsymbol{x}, \boldsymbol{u})) + L_\tau(\Phi(\boldsymbol{x}, \boldsymbol{u})) - L_{\tau-1}(\Phi(\boldsymbol{x}, \boldsymbol{u})).$$

By choosing $\tau = \tau(\epsilon)$ so that, $L_\tau(\boldsymbol{x}) - L_{\tau-1}(\boldsymbol{x}) < \epsilon$, we can ensure,

$$L_\tau(\boldsymbol{x}) - L_{\tau-1}(\Phi(\boldsymbol{x}, \boldsymbol{u})) \leq L_\tau(\boldsymbol{x}) - L_\tau(\Phi(\boldsymbol{x}, \boldsymbol{u})) + \epsilon \tag{17}$$

Plugging these inequalities together, introducing a telescopic sum, and manipulating the order of variables slightly (shown in Appendix C.4), we have the following relation:

$$(A) \leq \epsilon + \lim_{T \to \infty} \frac{1}{T} \sum_{t=0}^{T-1} \mathbb{E}\left[L_\tau(\boldsymbol{X}(t+1)) - L_\tau(\Phi(\boldsymbol{X}(t), \boldsymbol{u}(t))) - \lambda \cdot \left[\boldsymbol{X}(t+1) - \Phi(\boldsymbol{X}(t), \boldsymbol{u}(t))\right]\right].$$

$$\tag{18}$$

$L_\tau(\mathbf{x})$ is Lipschitz in $\mathbf{x}$ for any $\tau$. Since, $\mathbb{E}[\boldsymbol{X}(t+1)|\boldsymbol{X}(t), \boldsymbol{U}(t)] = \Phi(\boldsymbol{X}(t), \boldsymbol{U}(t))$, Lemma 1 of Gast et al. (2023b) implies:

$$\|\mathbb{E}[\boldsymbol{X}(t+1)|\boldsymbol{X}(t), \boldsymbol{U}(t)] - \Phi(\boldsymbol{X}(t), \boldsymbol{u}(t))\|_1 \leq \frac{\sqrt{|\mathcal{S}|}}{\sqrt{N}} + \frac{C_\Phi(\alpha N - \lfloor \alpha N \rfloor)}{N},$$

where $C_\Phi$ is the Lipschitz constant of the map $\Phi$. The Lipschitz constant of $L_\tau(\cdot)$ in the $l1$ norm is $\frac{k}{\rho_k}$ and $C_\lambda$ is equal to $\|\lambda\|_\infty$. Passing to the limit $\tau \to \infty$ we have:

$$V_{\text{opt}}^N(\mathbf{x}) - V_{\text{LP},\tau}^{(N)}(\mathbf{x}) \leq \epsilon + \left(\frac{k}{\rho} + C_\lambda\right)\left(\frac{\sqrt{|\mathcal{S}|}}{\sqrt{N}} + \frac{C_\Phi(\alpha N - \lfloor \alpha N \rfloor)}{N}\right) + \frac{(\alpha N - \lfloor \alpha N \rfloor)}{N} \tag{19}$$

$\square$

## 5.2 Theorem 4.2

The proof Theorem 4.2 is more classical and follows the same line as the proof of the exponential asymptotic optimality of Gast et al. (2023a); Hong et al. (2024a). The first ingredient is Lemma D.1 that shows that, by non-degeneracy, there exists a neighborhood of $\boldsymbol{x}^*$ such that $\mu_\tau(\boldsymbol{x}^*)$ is locally linear around $\boldsymbol{x}^*$. The second ingredient is Lemma D.2 that shows that $\boldsymbol{X}(t)$ is concentrated around $\boldsymbol{x}^*$ as $t \to \infty$, i.e., for all $\epsilon > 0$, there exists $C > 0$ such that $\lim_{t \to \infty} \Pr[\|\boldsymbol{X}(t) - \boldsymbol{x}^*\| \geq \varepsilon] \leq e^{-CN}$. Combining the two points imply the result. For more details, see Appendix D.

## 6 NUMERICAL ILLUSTRATIONS

In this section, we illustrate numerically the performance of the LP-update policy. We show that it performs very well in practice and outperform classical heuristics in most cases. We choose to compare against two heuristics of the literature: the LP-priority policy of Gast et al. (2023a) and the follow the virtual advice policy (FTVA) of Hong et al. (2023). The LP-priority policy computes a solution to (5) and uses it to compute a priority order on states, arms are then pulled according to this priority order till the budget is exhausted. FTVA constructs a virtual collection of $N$ arms associated with each arm which follow the solution to (5). If a virtual arm is pulled, the corresponding real arm is pulled if the budget allows it, otherwise the real arm is left alone. When the virtual and real arm choose the same actions, they evolve identically, otherwise they evolve independently.

We choose these two heuristics because they are natural and simple to implement and do not rely on hard-to-tune hyper-parameters (further discussion for our choice can be found in Appendix F). All parameters of the examples are provided in Appendix F. All codes are provided in supplementary material. In the future, a Github link will be provided.

**Comparison on representative examples**  For our first comparison, we consider three representative examples: a randomly generated example (in dimension 8), the main example used in Hong et al. (2023) and described in their Appendix G.2 (abbreviated Example Hong et al. (2023) in the following), and Example 2 of Figure 7.4 of Yan (2022) (abbreviated Example Yan (2022)). We report the result in Figure 1. We observe that in all three cases, both the LP-update and the FTVA policies are asymptotically optimal, but the LP-update policy outperforms FTVA substantially. The situation for LP-priority is quite different: as shown in Gast et al. (2023a), in dimension 8, the LP-priority policy is asymptotically optimal for most of the randomly generated examples. This is the case for example (a) of Figure 1 for which LP-update and LP-priority give an essentially equivalent performance and are essentially almost optimal. Note, the authors of Yan (2022) provide the numerical value of the optimal policy $N$ for Example Yan (2022), which can be done because $|\mathcal{S}| = 3$. We observe that the LP-update is very close to this value.

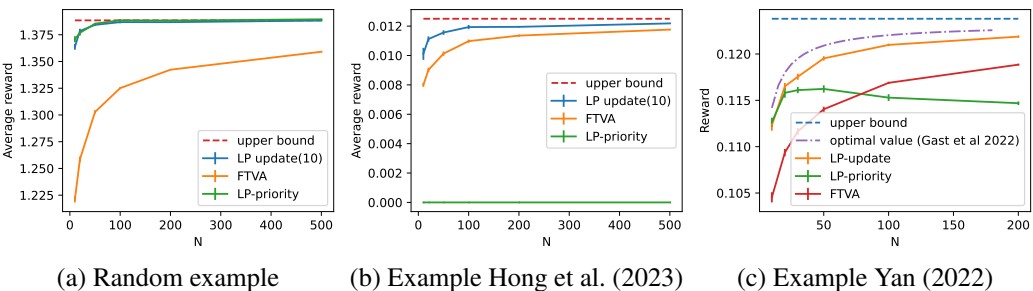

(a) Random example      (b) Example Hong et al. (2023)      (c) Example Yan (2022)

Figure 1: Performance as a function of $N$

**System dynamics:**  To explore the difference of performance between the LP-update policy and FTVA, we study the dynamics as a function of time of the different policies for the first two examples of Figure 1. We plot in Figure 2 the evolution of the system as a function of time (for a single trajectory). We observe that if the distance between $X_t$ and $x^*$ are similar for LP-update and FTVA, the rotated cost is much smaller for LP-update, which explains its good performance. To explore

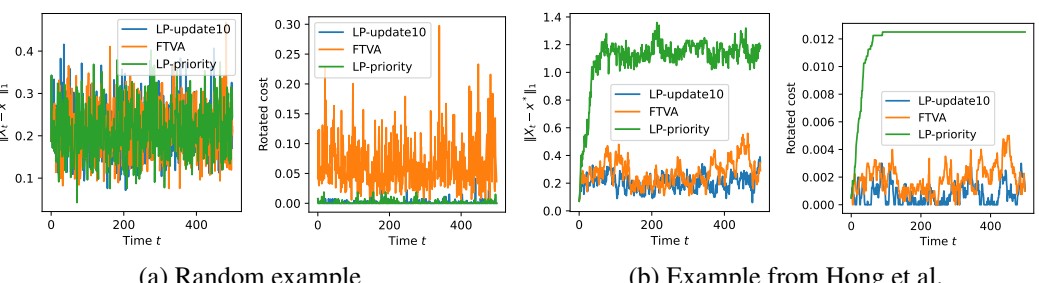

(a) Random example          (b) Example from Hong et al.

Figure 2: Distance $\|X_t - x^*\|$ and rotated cost as a function of time for $N = 100$.

the system dynamics in more details, we focus on Example Yan (2022) and study the behavior of

the LP-update, FTVA and LP-priority policy. In Figures 3(a,b,c), we present a trajectory of each of the three policies: each orange point corresponds to a value of $X_t \in \Delta_{\mathcal{S}}$ (as this example is in dimension $|\mathcal{S}| = 3$, the simplex $\Delta_{\mathcal{S}}$ can be represented as a triangle). This example has an unstable fixed point (Assumption 3 is not satisfied). Both LP-update and FTVA concentrate their behavior around this fixed point but the LP-priority policy exhibits two modes. When concentrating on the rotated cost (Figure 3(d)), we observe that it is much smaller for LP-update when compared to FTVA. This explains why LP-update performs better as shown in Figure 1(c).

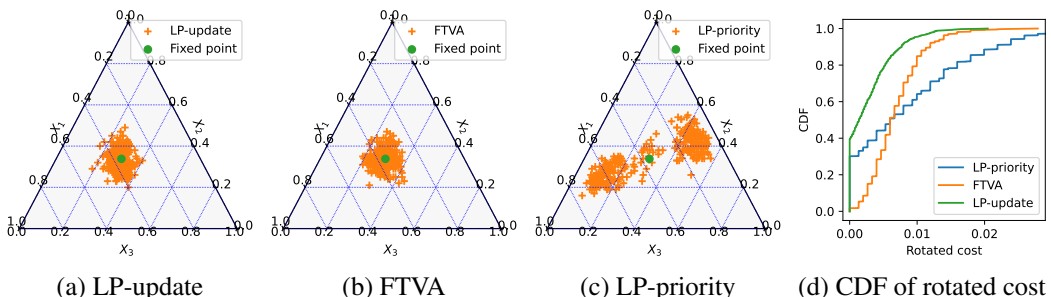

| (a) LP-update | (b) FTVA | (c) LP-priority | (d) CDF of rotated cost |

Figure 3: Example 2 from Gast et al. 22. Simulation for $N = 100$.

**Influence of parameters**  In Figure 4, we study the influence of different parameters on the performance of LP-update. In each case, we generated 20 examples and plot the average "normalized" performance among these 20 examples. "normalized" here means we divide the performance of the policy by the value of the LP problem (5). The first plot 4(a) shows that the influence of the parameter $\tau$ is marginal (the curves are not distinguishable for $\tau \in \{3, 5, 10\}$). Plot 4(b) indicates that the sub-optimality gap of LP-update is not too sensitive to the state space size unlike FTVA which degrades when $|\mathcal{S}|$ is large, probably because there are fewer synchronized arms. The last plot 4(c) studies the performance as a function of the budget parameter, $\alpha$.

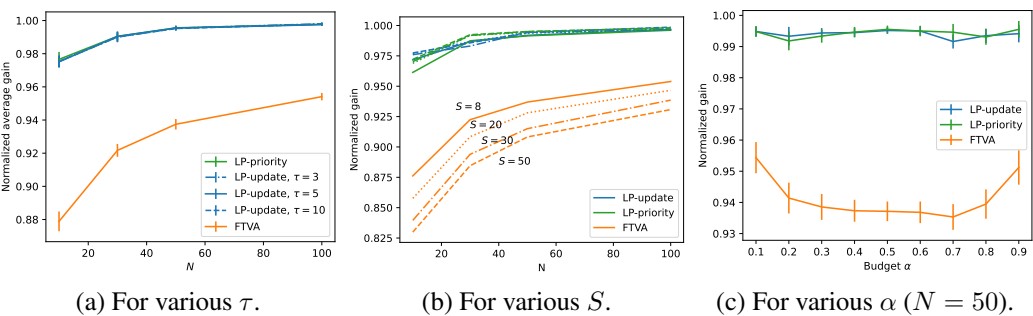

| (a) For various $\tau$. | (b) For various $S$. | (c) For various $\alpha$ ($N = 50$). |

Figure 4: Comparison of the gains as a function of some parameters.

## 7 CONCLUSION

In this paper, we study the problem of constructing an efficient policy for restless multi-armed bandit for the average reward case. We show that under quite general condition, a simple model-predictive control algorithm provides a solution that achieves both the best known upper bound up to know ($O(1/\sqrt{N})$ or $\exp(-\Omega N)$ for stable non-degenerate problems), but also works very efficiently in practice. Our paper provides the first analysis of this policy for the average reward criterion. Our paper uses a novel framework based on dissipativity that helps up to make a crisp connection between the finite- and the infinite-horizon problems and we are the first to us it in this context. We believe that this framework is what makes our analysis simple and easily generalizable to other settings. As an example, we discuss in Appendix B.2 potential generalization to the multi-action multi-constraint bandits, that is an interesting direction for future work.

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

## A   ADDITIONAL LITERATURE REVIEW

The study of bandit problems dates back at least to the 1930's in the form of sequential design of experiments, (Thompson, 1933), (Wald, 1947). However, the more modern perspective of Markovian bandits in terms of MDPs is largely credited to the monograph by Bellman et al. (1957). This monograph brought to light the combinatorial difficulty of this problem even over a finite horizon. The *rested* variant of the multi-armed bandit problem was famously resolved through Gittins (1979) which proposed a remarkably simple structure to the solution by allotting indices to the states of the bandit arms.

In his seminal work Whittle (1988) generalized the problem to the *restless multi-armed bandit problem* in its modern form and conjectured that under a condition known as indexability, an index policy is asymptotically optimal (in the number of arms) for the RMAB problem. Weber & Weiss (1990) showed a counter example to this conjecture, further going on to show in the same work that the conjecture did hold under a *global attractor* (UGAP assumption) and ergodicity condition of a single-arm. This result was generalized to the multi-action setting by Hodge & Glazebrook (2015). On the other hand, Verloop (2016); Gast et al. (2023b) generalized the set of index policies by constructing a set of priority policies known as LP priority index while maintaining the UGAP assumption. More surprisingly, Gast et al. (2023a;b) showed these priority policies are in fact exponentially close to being optimal, under three conditions: uniform global attractor property (UGAP), *non-degeneracy* of the fixed point and global exponential stability Gast et al. (2023a). The restriction of indexability is removed in Gast et al. (2023b) and the condition of UGAP was first removed Hong et al. (2024a).

Gast et al. (2023a;b) observed that under the conditions of non-degeneracy, there exists a local neighborhood around the fixed point where the priority policy set is affine. This is the key observation which leads to the exponentially small error bounds under simple local stability conditions. This observation around the fixed point drove several works (Hong et al., 2023; 2024a;b; Goldsztajn & Avrachenkov, 2024; Yan, 2024) to shift their perspective towards policies that look to steer the dynamical system towards the optimal fixed point. Hong et al. (2023) created a virtual system that was driven to the fixed point. To transform this into an asymptotically optimal policy, they required the real system to synchronize with the virtual system, hence, they developed the *synchronization assumption*. Unlike Hong et al. (2023) which passively allowed arms to align with each other eventually, Hong et al. (2024a) used a *two set policy* to actively align non aligned arms to the fixed point. In contrast to this form of control, Yan (2024) designed the *align and steer* policy to steer the mean field dynamical control system towards the fixed point. Yan (2024)'s policy is asymptotically optimal under the assumptions of *controllability* of the dynamical system to more directly move the state of the system towards the fixed point over time. In parallel to this work, Goldsztajn & Avrachenkov (2024) used a similar idea of mean field dynamical control for weakly coupled systems under a *relaxed but generalized constraint set* if the single arm process is unichain and aperiodic.

In contrast to these approaches our algorithm is not explicitly designed at the outset to steer the corresponding dynamical system towards a fixed point. Rather, we show that the model predictive control algorithm is designed to solve an equivalent *rotated cost minimization* problem. In doing so, it produces policies that are close to optimal to the infinite horizon average reward problem. We thus produce a policy that is easily generalizable to the multi-action, general constraint setting *without relaxing the constraints* using ideas from Whittle's relaxation. Further, when the fixed point is unique, minimizing the rotated cost problem will coincide with steering the dynamical control system towards the fixed point, allowing us to recover exponentially close asymptotic optimality bounds.

There are also a lot of recent papers on RMABs for the finite-horzon reward criteria Hu & Frazier (2017); Zayas-Caban et al. (2019); Brown & Smith (2020); Ghosh et al. (2022); Zhang & Frazier (2021); Gast et al. (2023a), or the infinite-horizon discounted reward Zhang & Frazier (2022); Ghosh et al. (2022). Our results are close in spirit to those since our policy shows how to use a finite-horizon policy to construct a policy that is asymptotically optimal for the infinite-horizon rewards. Finally, along the same lines, it is worth noting that a recent result Yan et al. (2024), developed in parallel to our own showed an order $\mathcal{O}(1/N)$ result for the *degenerate finite time horizon problem* using a diffusion approximation instead of a mean field approximation. Unfortunately, this finite horizon result cannot immediately be compared to the infinite horizon average reward result primarily because unlike the mean field solution which can directly be translated as the solution to (5) the diffusion dynamic has no immediate corresponding equivalent.

## B ALGORITHM DETAILS

### B.1 ROUNDING PROCEDURE

The rounding procedure is composed of two steps:

1. First, given $\boldsymbol{u} = \mu_\tau(\boldsymbol{x}(\mathbf{S}(t)))$, we need to construct a vector $\boldsymbol{U}^N$ that is *as close as possible* to $\boldsymbol{u}$ while satisfying the constraints that $\sum_i u_i \leq \alpha$ and $N\boldsymbol{U}^N$ is an integer.

2. Second, we use $\boldsymbol{U}^N$ to construct a feasible sequence of actions $\mathbf{A}(t) \in \{0,1\}^N$ that we can apply to $\mathbf{S}(t)$.

The second step is quite easy: as $Nx_i$ and $NU_i^N$ are integers for all $i$, we can construct a feasible sequence of action by applying the action 1 to $NU_i^N$ arms that in state $i$ and action 0 to the $N(x_i - U_i^N)$ others. The first step is more complicated because we want to construct a $U^i$ that is *as close as possible* as $u_i$. Before giving our procedure, let us illustrate it through two examples of $(N\boldsymbol{x}, N\boldsymbol{u})$ for $\alpha = 0.5$:

| $N\boldsymbol{x}$ | $N\boldsymbol{u}$ | Difficulty | Possible $\boldsymbol{U}^N$ by Algorithm 2 |
|---|---|---|---|
| $(10, 10, 10, 9)$ | $(10, 9.5, 0, 0)$ | $\|u\|_1 \geq \alpha$ | $(10, 9, 0, 0)$ |
| $(10, 10, 10, 9)$ | $(10, 5.7, 0.2, 0)$ | We need to randomize to be as close as possible to $\boldsymbol{u}$ | $(10, 6, 0, 0)$ with proba 0.7
$(10, 5, 1, 0)$ with proba 0.2
$(10, 5, 0, 0)$ with proba 0.1 |
| $(10, 10, 10, 9)$ | $(10, 4.9, 4.6, 0)$ | $\|u\|_1 \geq \alpha$ and we need to randomize | $(10, 5, 4, 0)$ with proba 0.4
$(10, 4, 5, 0)$ with proba 0.6 |

This leads us to construct Algorithm 2. This algorithm first construct a $\boldsymbol{v}$ that is as close as possible to $\boldsymbol{u}$ while satisfying $\|\boldsymbol{v}\| \leq \alpha$. There might be multiple choices for this first step and the algorithm can choose any. For instance, applied to the example such that $\boldsymbol{x} = (10, 10, 10, 9)$ and $\boldsymbol{u} = (10, 4.9, 4.6, 0)$, this algorithm would first construct a vector $\boldsymbol{v}$ that can be any convex combination of $(10, 4.4, 4.6, 0, 0)$ and $(10, 4.9, 4.1, 0, 0)$. For instance, if the algorithm chose the vector $\boldsymbol{v} = (10, 4.4, 4.6, 0)$, then the algorithm would produce $(10, 5, 4, 0)$ with probability 0.4 and $(10, 4, 5, 0)$ with probability 0.6. Once this is done, the algorithm outputs a $\boldsymbol{U}^N$ that satisfies the constraints and such that $\mathbb{E}[\boldsymbol{U}^N] = \boldsymbol{v}$. In some cases, there might be multiple distributions that work. The algorithm may output any. An efficient procedure to implemented this last step is provided in Section 5.2.3 of Ioannidis & Yeh (2016).

---

**Algorithm 2** Randomized rounding

---

**Input:** Integer $N$, vector $\boldsymbol{x} \in \Delta_{\mathcal{S}}$ such that $Nx_i$ is an integer for all $i$, vector $\boldsymbol{u} \in \mathcal{U}(\boldsymbol{x})$.

 Let $\boldsymbol{v}$ be (any) vector such that $\boldsymbol{v} \leq \boldsymbol{u}$ and $\|\boldsymbol{v}\|_1 = \min\left(\|\boldsymbol{u}\|_1, \frac{\lfloor \alpha N \rfloor}{N}\right)$.

 For all state $i$, let $z_i := Nv_i - \lfloor Nv_i \rfloor$ be the fractional part of $Nv_i$.

 Sample a sequence of $|\mathcal{S}|$ Bernoulli random variables $\boldsymbol{Z} = (Z_1 \ldots Z_{|\mathcal{S}|})$ such that $\mathbb{E}[Z_i] = z_i$ and $\sum_{i=1} Z_i \leq \lceil \sum_{i=1} z_i \rceil$.

**Output:** $(\lfloor Nv_i \rfloor + Z_i)_{i \in \mathcal{S}}$.

---

**Lemma B.1.** *Algorithm 2 outputs a random vector $\boldsymbol{U}^N$ such that $\boldsymbol{U}^N \in \mathcal{U}(\boldsymbol{x})$ and such that*

$$\|\mathbb{E}[\boldsymbol{U}^N] - \boldsymbol{u}\|_1 \leq \frac{\lfloor \alpha N \rfloor - \alpha N}{N}.$$

### B.2 GENERALIZATION TO MULTI-CONSTRAINTS MDPS

In this section, we give some clues on how and why our algorithm and results can be generalized to multi-action multi-constraints MDPs. Specifically, we make the following modifications to our model:

- Instead of restricting an action to be in $\{0, 1\}$, we can consider any finite action set $\mathcal{A} = \mathcal{A} = \{0 \ldots A - 1\}$.

- Instead of having a single constraint $\sum_n A_n(t) \leq \alpha N$, we consider that there are $K$ types of constraints and that when taking action $a$ in state $i$, this action consumes $D_{i,k}^a \geq 0$ of resource $k$. The action 0 is special and does not consume resources. We impose $K$ resource constraints:

$$\sum_{n=1}^{N} D_{S_n(t),k}^{A_n(t)} \leq C_k \qquad \forall k \in 1 \dots K.$$

As these new constraints are linear, one can define LP relaxation equivalence of (5) and (8), which leads to an LP-update adapted to this new setting. The only major modification of the algorithm concerns the randomized rounding and is where one would need to have an action 0 in order to guarantee the feasibility of the solution.

- The generalization of Assumption 1 to the multi-action case is straightforward. Hence, one can prove an equivalent of Theorem 4.1 for multi-action multi-constrained MDPs with a rate of convergence of $O(1/\sqrt{N})$.

- To obtain a generalization of Theorem 4.2, there are two main difficulties

  1. The first is to redefine the notion of a non-degenerate fixed-point in order to replace Assumption 2 by one adapted to the multi-action multi-constraint case. To that end, we believe that the most appropriate notion is the one presented in Gast et al. (2024). It provides the notion of non-degeneracy that uses a linear map.

  2. The second is to provide an equivalent of the stability Assumption 3. To do so, we can again use the notion of non-degeneracy defined in Gast et al. (2024) that defines a linear map. The stability of this linear map should suffice to prove the theorem.

## C    PROOF OF THEOREM 4.1

### C.1    PART 1, PROPERTIES OF THE DYNAMICAL CONTROL PROBLEM 8:

We begin by formally defining the space of stationary policies $\bar{\Pi}$ as the set of all policies that map $\boldsymbol{x} \in \Delta_{\mathcal{S}}$ to an action $\boldsymbol{u} \in \mathcal{U}(\boldsymbol{x})$. Note, any policy in $\Pi^{(N)}$ must satisfy this condition. Now, for an initial distribution $\boldsymbol{x} \in \Delta_{\mathcal{S}}$ define the discounted infinite horizon reward problem,

$$V_\beta^\infty(\boldsymbol{x}) := \max_{\pi \in \bar{\Pi}} \sum_{t=0}^{\infty} \beta^t R(\boldsymbol{x}(t), \boldsymbol{u}(t)) \qquad (20)$$

$$\text{Subject to the dynamics (7)} \qquad (21)$$

where $x(t)$ denotes the trajectory induced by the policy $\pi$. Note, this limit is always well defined for $\beta < 1$ and the optimal policy exists (Puterman, 2014). The main idea of this section will be an exercise in taking appropriate limit sequences of the discount factor $\beta$ to define the gain and bias for the constrained average reward problem.

Before we proceed we introduce a little notation for convenience in writing down our proof. For any pair $(\boldsymbol{x}, \boldsymbol{u})$ define an equivalent $\{y_{s,a}\}_{s,a}$ with:

$$y_{s,1} := \boldsymbol{u}_s$$
$$y_{s,0} := \boldsymbol{x}_s - \boldsymbol{u}_s$$

Here $y_{s,a}(t)$ represents the fraction of arms in state $s$, taking action $a$ at time $t$. It is also convenient to introduce a concatenated reward vector $\mathbf{r} := [\mathbf{r}^0, \mathbf{r}^1]$ and a concatenated transition kernel $\mathbf{P} := [\mathbf{P}^0, \mathbf{P}^1]$. If $\mathbf{y} := \{y_{s,a}\}$ represents the $\mathcal{S} \times 2$ vector, it is not hard to check that $R(\boldsymbol{x}, \boldsymbol{u}) := \mathbf{r}^0 \cdot \boldsymbol{x} + (\mathbf{r}^1 - \mathbf{r}^0) \cdot \boldsymbol{u} = \mathbf{r} \cdot \mathbf{y}$ and $\Phi(\boldsymbol{x}, \boldsymbol{u}) := \mathbf{P}^0 \cdot \boldsymbol{x} + (\mathbf{P}^1 - \mathbf{P}^0) \cdot \boldsymbol{u} = \mathbf{P} \cdot \mathbf{y}$. We can rewrite the

discounted problem as follows:

$$V_\beta^\infty(\boldsymbol{x}) := \max_{\pi \in \bar{\Pi}} \sum_{t=0}^\infty \beta^t \mathbf{r} \cdot \mathbf{y}(t) \tag{22}$$

$$y_{s,0}(t) + y_{s,1}(t) = \boldsymbol{x}_s(t) \tag{23}$$

$$\sum_a y_{s,a}(t+1) = \sum_{s',a'} P_{s',s}^{a'} y_{s',a'}(t) \quad \forall\ t \tag{24}$$

s.t.

$$\sum_s y_{s,1}(t) \leq \alpha \quad \forall\ t \tag{25}$$

We now state the following lemma on the continuity of the value function in the state:

**Lemma C.1.** *Under assumption 1, for any $\beta \in (0,1]$ we have,*

$$V_\beta^\infty(\boldsymbol{x}) - V_\beta^\infty(\boldsymbol{x}') \leq \frac{k}{\rho_k} \|\boldsymbol{x} - \boldsymbol{x}'\|_1 \tag{26}$$

Note, this continuity result is independent of the discount factor $\beta$. We postpone the proof to Appendix E. This will be critical to proving the main result of the subsection below.

**Lemma C.2.** *Consider the infinite horizon average reward problem under the synchronization assumption. There exists a constant gain and a bias function from $\Delta_\mathcal{S} \to \mathbb{R}$ denoted by $g^\star$ and $h^\star(\cdot)$ respectively defined by:*

$$g^\star := \lim_{i \to \infty} (1 - \beta_i) V_{\beta_i}^\infty(\boldsymbol{x})$$

*for an appropriate sequence $\beta_i \to 1$ as $i \to \infty$. For the same sequence of $\beta_i$ we can define the Lipschitz continuous bias function,*

$$h^\star(\boldsymbol{x}) := \lim_{i \to \infty} \sum_{t=0}^\infty \beta_i^t R(\boldsymbol{x}(t), \boldsymbol{u}(t)) - (1 - \beta_i)^{-1} g^\star$$

*Furthermore, they satisfy the following fixed point equations,*

$$g^\star + h^\star(\boldsymbol{x}) = \max_{\boldsymbol{u} \in \mathcal{U}(\boldsymbol{x})} R(\boldsymbol{x}(t), \boldsymbol{u}(t)) + h^\star(\Phi(\boldsymbol{x}, \boldsymbol{u})) \tag{27}$$

*Proof.* Now note, $\mathbf{y} \cdot \mathbf{1} = 1$, where $\mathbf{1}$ is the all $1's$ vector. We will overload the notation slightly by letting $\mathbf{y}(t)$ be the optimal trajectory taken by the optimal policy $\pi_\beta^*$ for 20. Hence, for any constant $g$ we have:

$$V_\beta^\infty(\boldsymbol{x}) = (1 - \beta)^{-1} g + \sum_{t=0}^\infty \beta^t [\mathbf{r} - g\mathbf{1}] \cdot \mathbf{y}(t) \tag{28}$$

One can check that $V_\beta^\infty(\boldsymbol{x})$ satisfies the following *Bellman equation*,

$$V_\beta^\infty(\boldsymbol{x}) = \mathbf{r} \cdot \mathbf{y}(t) + \beta V_\beta^\infty(\mathbf{P} \cdot \mathbf{y}) \tag{29}$$

Now note, given $\mathbf{y}(t)$ at time $t$, $\boldsymbol{x}(t+1)$ is given by a linear (hence, continuous) map $\boldsymbol{x}(t+1) := \mathbf{P} \cdot \mathbf{y}(t)$. Further note, given $\pi_\beta^*$, $\mathbf{y}(t)$ is upper hemicontinuous with respect to $\boldsymbol{x}(t)$. Let $\boldsymbol{H}_\beta$ denote the map induced by $\pi_\beta^*$ from $\boldsymbol{x}$ to $\mathbf{y}$, then $\boldsymbol{H}_\beta(\boldsymbol{x}(t)) = \mathbf{y}(t)$. Combining the two maps, we have $\mathbf{P}\boldsymbol{H}_\beta : \Delta_\mathcal{S} \to \Delta_\mathcal{S}$. Since, $\Delta_\mathcal{S}$ is closed and bounded, by Kakutani's fixed point theorem there exists atleast one fixed point $\boldsymbol{x}_\beta := \mathbf{P}\boldsymbol{H}_\beta \cdot \boldsymbol{x}_\beta$ if $\mathbf{y}_\beta$ is the corresponding value for $\boldsymbol{x}_\beta$ we have $\boldsymbol{x}_\beta := \mathbf{P} \cdot \mathbf{y}_\beta$. Combining these observations allows us to write,

$$V_\beta^\infty(\boldsymbol{x}_\beta) = \mathbf{r} \cdot \mathbf{y}_\beta + \beta V_\beta^\infty(\boldsymbol{x}_\beta)$$

We will set $g_\beta := \mathbf{r} \cdot \mathbf{y}_\beta$ and designate this value as the *gain* of our problem. It then follows that $V_\beta^\infty(\boldsymbol{x}_\beta) = (1 - \beta)^{-1} g_\beta$. Now, we can define a *bias function* $h_\beta(\cdot)$ as the remaining terms of the equation (28), concretely,

$$h_\beta(\boldsymbol{x}) = \sum_{t=0}^\infty \beta^t [\mathbf{r} - g_\beta \mathbf{1}] \cdot \mathbf{y}(t) \tag{30}$$

and this definition of $h_\beta(\cdot)$ is well defined since the sum is bounded. It now follows that,

$$V_\beta^\infty(\boldsymbol{x}) = (1-\beta)^{-1}g_\beta + h_\beta(\boldsymbol{x}) \tag{31}$$

More importantly, by plugging these definitions into (29) and making the dependence of $\mathbf{y}$ on the maximal policy, we obtain the following recursion equation:

$$g_\beta + h_\beta(\boldsymbol{x}) = \max_{\mathbf{y}:\{\mathbf{y}_1 \in \mathcal{U}(\boldsymbol{x})\}} g_\beta + \beta h_\beta(\mathbf{P}\cdot\mathbf{y}) \tag{32}$$

Further, due to the Lipschitz continuity, Lemma C.1 of $V_\beta^\infty(\boldsymbol{x})$ with respect to $\boldsymbol{x}$ we have, for any discount factor $\beta_i < 1$,

$$(1-\beta_i)\|V_{\beta_i}^\infty(\boldsymbol{x}) - V_{\beta_i}^\infty(\boldsymbol{x}')\|_1 \le \frac{k(1-\beta_i)}{\rho_k}\|\boldsymbol{x}-\boldsymbol{x}'\|_1$$

. Hence, by choosing a sequence $\beta_i \uparrow 1$ as $i \to \infty$ we obtain:

$$\lim_{i\to\infty}(1-\beta_i)\|V_{\beta_i}^\infty(\boldsymbol{x}) - V_{\beta_i}^\infty(\boldsymbol{x}')\|_1 \to 0$$

. From Puterman (2014) we know there exists a sequence $i \to \infty$ such that $(1-\beta_i)V_{\beta_i}^\infty(\boldsymbol{x}) \to \lim_{T\uparrow\infty}\frac{W_T(\boldsymbol{x})}{T}$. In particular, this implies that $(1-\beta_i)V_{\beta_i}^\infty(\boldsymbol{x})$ converges to a constant value which we shall denote by $g^\star$. This gives us the first result. Critically, by noting that $g_{\beta_i} := (1-\beta_i)V_{\beta_i}^\infty(\boldsymbol{x}_{\beta_i})$ we see that $g^\star$ must be the limit point of $R(\boldsymbol{x}_{\beta_i}, \boldsymbol{u}_{\beta_i}) =: \mathbf{r}\cdot\mathbf{y}_{\beta_i} =: g_{\beta_i}$, hence, it is the solution to (5).

Next, once again leveraging Lemma C.1 we have, for any $\beta < 1$,

$$\|V_\beta^\infty(\boldsymbol{x}) - (1-\beta)^{-1}g_\beta\|_1 := \|V_\beta^\infty(\boldsymbol{x}) - V_\beta^\infty(\boldsymbol{x}_\beta)\|_1 \le \frac{k}{\rho_k}\|\boldsymbol{x}-\boldsymbol{x}_\beta\|_1$$

Plugging these results into (31) we see that,

$$\|h_\beta(\boldsymbol{x})\|_1 \le \frac{k}{\rho_k}\|\boldsymbol{x}-\boldsymbol{x}_\beta\|_1 \tag{33}$$

is bounded by a constant for all $\beta < 1$. By the same token $h_\beta$ is Lipschitz with constant $\frac{k}{\rho_k}$. Choosing the same sequence $\beta_i$ in (30) and passing to the limit we obtain :

$$h^\star(\boldsymbol{x}) := \lim_{i\to\infty} h_{\beta_i}(\boldsymbol{x}) = \lim_{i\to\infty}\sum_{t=0}^\infty \beta_i^t[\mathbf{r} - g_{\beta_i}\mathbf{1}]\cdot\mathbf{y}(t) = \lim_{i\to\infty}\sum_{t=0}^\infty \beta_i^t\mathbf{r}\cdot\mathbf{y}(t) - (1-\beta_i)^{-1}g^\star$$

Thus, we have defined both the gain and bias $g^\star$ and $h^\star(\cdot)$ for our *deterministic average reward problem*. Recall, $R(\boldsymbol{x},\boldsymbol{u}) := \mathbf{r}^0\cdot\boldsymbol{x} + (\mathbf{r}^1-\mathbf{r}^0)\cdot\boldsymbol{u} = \mathbf{r}\cdot\mathbf{y}$ giving us the second result:

$$h^\star(\boldsymbol{x}) = \lim_{i\to\infty}\sum_{t=0}^\infty \beta_i^t R(\boldsymbol{x}(t),\boldsymbol{u}(t)) - (1-\beta_i)^{-1}g^\star \tag{34}$$

Further, this sequence $\beta_i \uparrow 1$ in (32) yields the following fixed point equation:

$$g^\star + h^\star(\mathbf{x}) = \max_{\boldsymbol{u}\in\mathcal{U}(\boldsymbol{x})} R(\boldsymbol{x}(t),\boldsymbol{u}(t)) + h^\star(\Phi(\boldsymbol{x}(t),\boldsymbol{u}(t)))$$

which completes the proof. □

## C.2 PART 2: DISSIPATIVITY

**Lemma C.3.** *Let* $\tilde{l}(\boldsymbol{x},\boldsymbol{u}) = g^* - R(\boldsymbol{x},\boldsymbol{u}) + \lambda\cdot\boldsymbol{x} - \lambda\cdot\Phi(\boldsymbol{x},\boldsymbol{u})$. *Then:*

- $\tilde{l}(\boldsymbol{x},\boldsymbol{u}) \ge 0$ *for all* $\boldsymbol{x}\in\Delta_\mathcal{S}$ *and* $\boldsymbol{u}\in\mathcal{U}(\boldsymbol{x})$.

- *If* $(\boldsymbol{x}^*,\boldsymbol{u}^*)$ *is a solution to* (5), *then* $\tilde{l}(\boldsymbol{x}^*,\boldsymbol{u}^*) = 0$.

*This shows that our problem is dissipative.*

*Proof.* Recall that $\lambda$ is the optimal dual variable of the constraint $\mathbf{x} = \mathbf{P}^0 \cdot \mathbf{x} + (\mathbf{P}^1 - \mathbf{P}^0) \cdot \boldsymbol{u}$ of problem (5). By strong duality and Lemma C.2, this implies that

$$g^* = \max_{\boldsymbol{x}, \boldsymbol{u}} R(\boldsymbol{x}, \boldsymbol{u}) + \lambda \cdot (\mathbf{P}^0 \cdot \boldsymbol{x} + (\mathbf{P}^1 - \mathbf{P}^0) \cdot \boldsymbol{u}) - \lambda \cdot \boldsymbol{x} \qquad \text{Subject to:} \quad \boldsymbol{u} \in \mathcal{U}(\mathbf{x}).$$

Recall that $\Phi(\boldsymbol{x}, \boldsymbol{u}) = \mathbf{P}^0 \cdot \boldsymbol{x} + (\mathbf{P}^1 - \mathbf{P}^0) \cdot \boldsymbol{u}$. This implies that

$$0 = \min_{\boldsymbol{x}, \boldsymbol{u}} \tilde{l}(\boldsymbol{x}, \boldsymbol{u}) \qquad \text{Subject to: } \boldsymbol{u} \in \mathcal{U}(\boldsymbol{x}),$$

which implies the results of the lemma. $\qquad \square$

### C.3  PART 3: MPC IS OPTIMAL FOR THE DETERMINISTIC PROBLEM

Recall that $L_\tau(\mathbf{x}) := \min \sum_{t=0}^{\tau-1} \tilde{l}(\boldsymbol{x}(t), \boldsymbol{u}(t))$ with $\boldsymbol{x}(0) = \mathbf{x}$.

**Lemma C.4.** *For all $\boldsymbol{x} \in \Delta_{\mathcal{S}}$, for any $\epsilon > 0$, there exists $t$ such that:*

$$L_t(\boldsymbol{x}) - L_{t-1}(\Phi(\boldsymbol{x}, \boldsymbol{u})) < \epsilon.$$

*where $\boldsymbol{u} \in \mathcal{U}(\boldsymbol{x})$ is $\mu_t(\boldsymbol{x})$.*

*Proof.* Recall that the objective function of the optimization problems $W_\tau$ and $L_\tau$ are:

$$\sum_{t=0}^{T-1} \tilde{l}(\boldsymbol{x}(t), \boldsymbol{u}(t)) \qquad \text{For } L_\tau$$

$$\sum_{t=0}^{T-1} R(\boldsymbol{x}(t), \boldsymbol{u}(t)) + \lambda(\boldsymbol{x}(\tau)). \qquad \text{For } W_\tau$$

By Lemma C.3, the rotated cost is $\tilde{l}(\mathbf{x}, \boldsymbol{u}) = g^* - R(\boldsymbol{x}, \boldsymbol{u}) + \lambda(\boldsymbol{x}) - \lambda(\Phi(\boldsymbol{x}, \boldsymbol{u}))$. As any valid control of $L_\tau$ and $W_\tau$ satisfy $\boldsymbol{x}(t+1) = \Phi(\boldsymbol{x}(t), \boldsymbol{u}(t))$, the objective of $L_\tau$ can be rewritten as:

$$\sum_{t=0}^{\tau-1} \tilde{l}(\boldsymbol{x}(t), \boldsymbol{u}(t)) = \sum_{t=0}^{\tau-1} g^* - R(\boldsymbol{x}(t), \boldsymbol{u}(t)) + \lambda(\boldsymbol{x}(t)) - \lambda(\Phi(\boldsymbol{x}(t), \boldsymbol{u}(t)))$$

$$= \lambda(\boldsymbol{x}(0)) + \tau g^* - \sum_{t=0}^{T-1} R(\boldsymbol{x}(t), \boldsymbol{u}(t)) - \lambda(\boldsymbol{x}(\tau)),$$

As the last two terms correspond to the objective function for $W_\tau$, this shows that $L_\tau(\boldsymbol{x}) = \tau g^\star + \lambda \cdot \boldsymbol{x} - W_\tau(\boldsymbol{x})$. Clearly, the two objectives are equivalent and $\mu_\tau(\boldsymbol{x}) = \boldsymbol{u}(0)$. Now note, by Lemma C.2 we have,

$$\lim_{\tau \to \infty} L_\tau(\boldsymbol{x}) = \lambda \cdot \boldsymbol{x} + h^\star(\boldsymbol{x}) < \infty$$

is bounded. Due to dissipativity, $L_\tau(\mathbf{x})$ is monotone increasing in $\tau$. It follows that

$$L_\tau(\boldsymbol{x}) - L_{\tau-1}(\Phi(\boldsymbol{x}, \boldsymbol{u})) < \epsilon.$$

This completes the proof. $\qquad \square$

### C.4  PROOF OF THEOREM 4.1: COMPUTATION DETAILS

Here we detail the analysis of $(A)$ and how we go from (16) to (18) in the proof of Theorem 4.1. This term is equal to

$$\frac{1}{T} \sum_{t=0}^{T-1} \mathbb{E}\left[l(\boldsymbol{X}(t), \boldsymbol{u}(t))\right] = \frac{1}{T} \sum_{t=0}^{T-1} \mathbb{E}\left[\tilde{l}(\boldsymbol{X}(t), \boldsymbol{u}(t)) - \lambda \cdot \boldsymbol{X}(t) + \lambda \cdot \Phi(\boldsymbol{X}(t), \boldsymbol{u}(t))\right]$$

$$= \frac{1}{T} \sum_{t=0}^{T-1} \mathbb{E}\left[L_\tau(\boldsymbol{X}(t)) - L_{\tau-1}(\Phi(\boldsymbol{X}(t), \boldsymbol{u}(t)) - \lambda \cdot \boldsymbol{X}(t) + \lambda \cdot \Phi(\boldsymbol{X}(t), \boldsymbol{u}(t))\right]$$

$$\leq \frac{1}{T} \sum_{t=0}^{T-1} \mathbb{E}\left[L_\tau(\boldsymbol{X}(t)) - L_\tau(\Phi(\boldsymbol{X}(t), \boldsymbol{u}(t))) + \epsilon - \lambda \cdot \boldsymbol{X}(t) + \lambda \cdot \Phi(\boldsymbol{X}(t), \boldsymbol{u}(t))\right], \quad (35)$$

where we use the definition of the rotated cost for the first equality, the dynamic principle for the second line and the identity (17) for the last time.

$$\frac{1}{T} \sum_{t=0}^{T-1} [L_\tau(\boldsymbol{X}(t)) - \lambda \cdot \boldsymbol{X}(t)] = \frac{1}{T} \sum_{t=0}^{T-1} [L_\tau(\boldsymbol{X}(t+1)) - \lambda \cdot \boldsymbol{X}(t+1)]$$
$$+ \frac{1}{T} (L_\tau(\boldsymbol{X}(0)) - L_\tau(\boldsymbol{X}(T)) + \lambda \cdot \boldsymbol{X}(0) - \lambda \cdot \boldsymbol{X}(T))$$

When $T$ goes to infinity, the second line of the above result goes to 0. This shows that when we take the limit as $T$ goes to infinity, (35) is equal to

$$\epsilon + \lim_{T \to \infty} \frac{1}{T} \sum_{t=0}^{T-1} [L_\tau(\boldsymbol{X}(t+1)) - L_\tau(\Phi(\boldsymbol{X}(t), \boldsymbol{u}(t))) - \lambda \cdot \boldsymbol{X}(t+1) + \lambda \cdot \Phi(\boldsymbol{X}(t), \boldsymbol{u}(t))],$$

which shows (18)

## D  PROOF OF THEOREM 4.2

**Lemma D.1.** *Assume 2 and 3, then there exists a neighborhood $\mathcal{N}$ of $\boldsymbol{x}^*$ and a matrix $A$ such that $\mu_\tau(\boldsymbol{x}) = \mu_\tau(\boldsymbol{x}^*) + A(\boldsymbol{x} - \boldsymbol{x}^*)$ for all $\boldsymbol{x} \in \mathcal{N}$.*

*Proof.* Let us denote by $S^+ := \{i : u_i^* = x_i^*\}$ the set of states for which all the action 1 is taken for all arms in those state and by $S-+ := \{i : u_i^* = x_i^*\}$ the set of states for which all the action 0 is taken for all arms in those state. Recall that $i^*$ is the unique state such that $0 < u_i^* < x_i^*$ (which exists and is unique by Assumption 2). We define the function $f : \Delta_\mathcal{S} \to \mathcal{S}$ by:

$$f_i(x) = \begin{cases} x_i & \text{For all } i \in \mathcal{S}^+ \\ \alpha - \sum_{i \in \mathcal{S}^+} x_i & \text{For } i = i^* \\ 0 & \text{For all } i \in \mathcal{S}^-. \end{cases}$$

We claim that there exists a neighborhood $\mathcal{N}$ such that

1. For all $\boldsymbol{x} \in \mathcal{N}$, we have $f(\boldsymbol{x}) \in \mathcal{U}(\boldsymbol{x})$, which means that $f(\boldsymbol{x})$ is a feasible control for $\boldsymbol{x}$.

   *Proof.* We remark that by construction, one has $\sum_i f_i(\boldsymbol{x}) = \alpha$. Hence, $f(\boldsymbol{x})$ is a valid control if and only if $0 \le \boldsymbol{u} \le \boldsymbol{x}$. This is clearly true for all $i \ne i^*$. For $i^*$, it is true if $0 \le \alpha - \sum_{i\mathcal{S}^+} x_i \le x_i^*$ which holds in a neighborhood of $\boldsymbol{x}$ because of non-degeneracy (Assumption 2) that implies that $0 < u_i^* = \alpha - \sum_{i\mathcal{S}^+} x_i^* < x_i^*$.

2. There exists a neighborhood $\mathcal{N}'$ of $\boldsymbol{x}^*$ such that if $\boldsymbol{x}(0) \in \mathcal{N}'$, and we construct the sequence $\boldsymbol{x}(t)$ by setting $\boldsymbol{x}(t+1) = \Phi(\boldsymbol{x}(t), f(\boldsymbol{x}(t)))$, then $\boldsymbol{x}(t) \in \mathcal{N}$ for all $t$.

   *Proof.* We remark that $\boldsymbol{x}(t+1) = \boldsymbol{x}^* + (\boldsymbol{x}(t) - \boldsymbol{x}^*)P^*$, where $P^*$ is the matrix defined in Assumption 3. Indeed, one has:

   $$\Phi_j(\boldsymbol{x}, f(\boldsymbol{x})) = (\boldsymbol{x}P^0)_j + (f(\boldsymbol{x})(P^1 - P^0))_j$$
   $$= \sum_{i \in \mathcal{S}^+} x_i P_{ij}^1 + \sum_{i \in \mathcal{S}^-} x_i P_{ij}^0 + x_{i^*} P_{i^*,j}^0 + (\alpha - \sum_{i \in \mathcal{S}^+} x_i)(P_{i^*,j}^1 - P_{i^*,j}^0)$$
   $$= \sum_{i \in \mathcal{S}^- \cup \{i^*\}} x_i P_{ij}^0 + \sum_{i \in \mathcal{S}^+} x_i (P_{i,j}^1 - P_{i^*,j}^1 + P_{i^*,j}^0) + \alpha(P_{i^*,j}^1 - P_{i^*,j}^0)$$
   $$= (\boldsymbol{x}P^*)_j + \alpha(P_{i^*,j}^1 - P_{i^*,j}^0),$$

   where $P^*$ is the matrix defined in Assumption 3.

   Note that by construction, one has $\Phi(\boldsymbol{x}^*, f(\boldsymbol{x}^*)) = \boldsymbol{x}^*$. This implies that $\Phi_j(\boldsymbol{x}, f(\boldsymbol{x})) - \Phi_j(\boldsymbol{x}^*, f(\boldsymbol{x}^*)) = (\boldsymbol{x} - \boldsymbol{x}^*)P^*$. The result follows by Assumption 3 that imposes that the matrix $P^*$ is stable.

3. Let $Z(\boldsymbol{x})$ be the reward function collected by the control $f(\boldsymbol{x})$. $Z(\boldsymbol{x})$ is a linear function of $\boldsymbol{x}$ and $Z(\boldsymbol{x}^*) = \tau g^* + \lambda \cdot x^* = W_\tau(\boldsymbol{x}^*)$. As a result, $Z(\boldsymbol{x}) = W_\tau(\boldsymbol{x})$ for all $\boldsymbol{x} \in \mathcal{N}'$.

*Proof.* The linearity of $Z$ is a direct consequence of the linearity of $f$ and $\Phi$. The fact that $Z(\boldsymbol{x}^*) = \tau g^* + \lambda \cdot x^*$ is because $f(\boldsymbol{x}^*) = \boldsymbol{u}^*$. It is also the optimal value of $W_\tau(\boldsymbol{x}^*)$ by dissipativity. Last, as the function $W_t()$ is the solution of an LP, it is concave. Since $Z$ is linear and $0 < x_i < 1$ for all state $i$, it follows that the two functions must coincide.

$\square$

**Lemma D.2.** *Assume 1, and 4. Assume that $\alpha N$ is an integer. For any neighborhood $\mathcal{N}$ of $\boldsymbol{x}^*$ There exists a $C > 0$ such that if $\boldsymbol{X}(t)$ is a trajectory of the optimal control problem, then:*

$$\lim_{t \to \infty} \Pr[\boldsymbol{X}(t) \in \mathcal{N}] \le e^{-CN}.$$

*Proof.* We use two main steps:

1. For any initial condition $\boldsymbol{x}(0)$, let $\boldsymbol{x}_\tau(t+1) = \Phi(\boldsymbol{x}_\tau(t), \mu_\tau(\boldsymbol{x}_\tau(t)))$, then there exists $\tau$ and $T$ such that for any initial condition $\boldsymbol{x}(0)$ and any $t \ge T$, one has $\boldsymbol{x}_\tau(t) \in \mathcal{N}$.

   *Proof.* The result follows by dissipativity (which holds because of Assumption 1). Indeed, as $\boldsymbol{x}_\tau(\cdot)$ is an optimal trajectory, it must hold that $\lim_{\tau \to \infty, T \to \infty} \tilde{l}(\boldsymbol{x}_\tau(t), \mu_\tau(\boldsymbol{x}_\tau(t))) = 0$.

2. The map $\boldsymbol{x} \mapsto \mu_\tau(\boldsymbol{x})$ is Lipschitz-continuous in $\boldsymbol{x}$.

   *Proof.* By assumption 4, the control $\mu_\tau(\boldsymbol{x})$ is unique. The result then follows because $\mu_\tau(\boldsymbol{x})$ is the solution of a linear program parametrized by $\boldsymbol{x}$.

The lemma can then be proven by adapting the proof of Theorem 3 of Gast et al. (2023b) and in particular their Equation (17). This is proven by using Lemma 1 Gast et al. (2023b) that implies that $\Pr[\|\boldsymbol{X}(t+1) - \Phi(\boldsymbol{X}(t), \boldsymbol{U}(t))\| \ge \epsilon] \le e^{-C'N}$. $\square$

We are now ready to prove Theorem 4.2. We can follow the proof of Theorem 4.1 up to (18) that shows that $V_{\text{opt}}^N(\mathbf{x}) - V_{\text{LP},\tau}^{(N)}(\mathbf{x})$ is bounded[4] by

$$\epsilon + \lim_{\tau \to \infty} \frac{1}{\tau} \sum_{t=0}^{\tau-1} \mathbb{E}\left[L_\tau(\boldsymbol{X}(t+1)) - L_\tau(\Phi(\boldsymbol{X}(t), \boldsymbol{u}(t))) - \lambda \boldsymbol{X}(t+1) + \lambda \Phi(\boldsymbol{X}(t), \boldsymbol{u}(t))\right]$$

Let $g(\boldsymbol{x}) = L_\tau(\boldsymbol{x}) - \lambda \boldsymbol{x}$. By Lemma D.1, this function is linear on $\mathcal{N}$. Let us denote by $E(t)$ the event

$$E(t) := \{\boldsymbol{X}(t+1) \in \mathcal{N} \wedge \Phi(\boldsymbol{X}(t), \boldsymbol{u}(t)) \in \mathcal{N}\}.$$

Hence, this shows that:

$$\mathbb{E}\left[L_\tau(\boldsymbol{X}(t+1)) - L_\tau(\Phi(\boldsymbol{X}(t), \boldsymbol{u}(t))) - \lambda \boldsymbol{X}(t+1) + \lambda \Phi(\boldsymbol{X}(t), \boldsymbol{u}(t))\right]$$
$$= \mathbb{E}\left[g(\boldsymbol{X}(t+1)) - g(\Phi(\boldsymbol{X}(t), \boldsymbol{u}(t)))\right]$$
$$= \mathbb{E}\left[(g(\boldsymbol{X}(t+1)) - g(\Phi(\boldsymbol{X}(t), \boldsymbol{u}(t))))\mathbf{1}_{E(t)}\right] + \mathbb{E}\left[(g(\boldsymbol{X}(t+1)) - g(\Phi(\boldsymbol{X}(t), \boldsymbol{u}(t))))\mathbf{1}_{\bar{E}(t)}\right]$$

By linearity of the $g$ when $E(t)$ is true and by the fact that $\mathbb{E}[\boldsymbol{U}(t)] = \boldsymbol{u}(t)$ (Lemma B.1), the first term is equal to 0. Moreover, as $E(t)$ is true with probability at least $1 - 2e^{-CN}$, the second term is bounded by $C'e^{-CN}$ when $t$ is large.

This concludes the proof of Theorem 4.2.

---

[4]In this expression, the term $(\alpha N - \lfloor \alpha N \rfloor)/N$ is equal to 0 here because we assumed that $\alpha N$ is an integer.

# E    Proof of Lemma C.1

## Proof Outline

This section is dedicated to proving the Lipschitz property for the value function $V_\beta^\infty(\boldsymbol{x})$ in $\boldsymbol{x}$ under the ergodicity Assumption 1. For simplicity of exposition we will set $k = 1$, although it is not hard to extrapolate the proof for $k > 1$ and is left as an exercise for the reader.

The key (rather counter-intuitive) idea behind this proof is to rewrite the problem in terms of an $M$ component vector $\mathbf{s} \in \mathcal{S}^{(M)}$. As $M$ tends to infinity we will use the dense nature of rational numbers in the real line and continuity arguments to argue that the proof holds for any $\boldsymbol{x} \in \Delta_\mathcal{S}$.

Hence, to start with, we will assume that $\boldsymbol{x} \in \Delta_\mathcal{S}^{(M)}$, the set of all points on the simplex that can be represented by an $M$ component vector $\mathcal{S}^{(M)}$. In order to complete this proof we will need an intermediate result that verifies the Lipschitz property for all $\boldsymbol{x}, \boldsymbol{x}' \in \Delta_\mathcal{S}^M$ such that $\|\boldsymbol{x} - \boldsymbol{x}'\| \leq \frac{2}{M}$ i.e, they differ on at most two components.

## Proof of Lemma

Following the proof outline above let $\boldsymbol{x}, \boldsymbol{x}' \in \Delta_\mathcal{S}^{(M)}$. Hence, there exist unique (up to permutation) $M$ component vectors $\mathbf{s}(\boldsymbol{x}) := \{s_0, s_1 \ldots s_{M-1}\}$ and $\mathbf{s}'(\boldsymbol{x}) := \{s_0', s_1' \ldots s_{M-1}'\}$.

**Lemma E.1.** *Under assumption 1, for any $\beta \in (0, 1]$, let $\boldsymbol{x}, \boldsymbol{x}' \in \Delta_\mathcal{S}^{(M)}$ with $\|\boldsymbol{x} - \boldsymbol{x}'\|_1 \leq \frac{2}{M}$, then,*

$$V_\beta^\infty(\boldsymbol{x}) - V_\beta^\infty(\boldsymbol{x}') \leq \frac{1}{\rho}\|\boldsymbol{x} - \boldsymbol{x}'\|_1 \tag{36}$$

*Proof.* Let $\mathbf{s}(\boldsymbol{x}) := \{s_0, s_1 \ldots s_{M-1}\}$, WLOG we can assign $\mathbf{s}'(\boldsymbol{x}') := \{s_0', s_1, s_2 \ldots s_{M-1}\}$. Note, in this case if $s_0$ is the 1$^{\text{st}}$ component ($s_0 = 1$) then one can look at the fraction of arms in state 1, $\mathbf{x}(\mathbf{s})_1 := x_1$. Now if $s_0' = i \neq 1$, we have $x_1' = x_1 - \frac{1}{M}$ and $x_i' = x_i + \frac{1}{M}$, allowing us to conclude $\|\boldsymbol{x} - \boldsymbol{x}'\|_1 = \frac{2}{M}$. Note, for any $\boldsymbol{x}$, let $\mathbf{a}$ be an action vector *i.e.* $\mathbf{a} = \{a_0, a_1 \ldots a_{M-1}\}$. Since, $V_\beta^\infty(\boldsymbol{x})$ is always well defined for any $\beta < 1$, one can write the $Q$ function for state $\mathbf{s}$ and action $\mathbf{a}$ as follows:

$$Q(\mathbf{s}, \mathbf{a}) := \frac{1}{M}\sum_{n=0}^{M-1} R_{s_n}^{a_n} + \beta \sum_{\mathbf{s}'' \in \Delta_\mathcal{S}^{(M)}} V_\beta^\infty(\mathbf{x}(\mathbf{s}'')) \Pi_{n=0}^{M-1} P_{s_n, s_n''}^{a_n}$$

Now, there exists $\mathbf{a}^* := \{a_0, a_1 \ldots a_{M-1}\}$ such that $Q(\mathbf{s}, \mathbf{a}^*) = V_\beta^\infty(\mathbf{x}(\mathbf{s}))$. Suppose we pick $\mathbf{a}' := \{0, a_1, a_2 \ldots a_{M-1}\}$. Note, $\mathbf{a}'$ always satisfies the constraint on the action space (if needed by pulling one less arm). Hence,

$$V_\beta^\infty(\boldsymbol{x}) - V_\beta^\infty(\boldsymbol{x}') \leq Q(\mathbf{s}, \mathbf{a}^*) - Q(\mathbf{s}', \mathbf{a}')$$

$$= \frac{R_{s_0}^{a_0} - R_{s_0'}^0}{M} + \sum_{\mathbf{s}'' \in \mathcal{S}^{(M)}} V_\beta^\infty(\mathbf{x}(\mathbf{s}'')) \Pi_{n=1}^{M-1} P_{s_n, s_n''}^{a_n} \left( P_{s_0, s_0''}^{a_0} - P_{s_0', s_0''}^0 \right)$$

$$= \frac{R_{s_0}^{a_0} - R_{s_0'}^0}{M} +$$

$$\sum_{s_1'' \ldots s_{M-1}''} \Pi_{n=1}^{M-1} P_{s_n, s_n''}^{a_n} \left[ \sum_{i \in \mathcal{S}} V_\beta^\infty(\mathbf{x}(\{i, s_1'' \ldots s_{M-1}''\}))(P_{s_n, i}^{a_n} - P_{s_n', i}^0) \right]$$

Note, $0 \le R_{s_0}^{a_0} \le 1$, so the first term is bounded above by $\frac{1}{M}$. We will focus on the second term, to this end let $\rho(i) = \min\{P_{s_0,i}^{a_0}, P_{s_0',i}^0\}$.

$$\left[\sum_{i \in \mathcal{S}} V_\beta^\infty(\mathbf{x}(\{i, s_1'' \dots s_{M-1}''\}))(P_{s_n,i}^{a_n} - P_{s_n',i}^0)\right]$$

$$= \sum_{i \in \mathcal{S}} V_\beta^\infty(\mathbf{x}(\{i, s_1'' \dots s_{M-1}''\}))(P_{s_0,i}^{a_0} - \rho(i)) - \sum_{i \in \mathcal{S}} V_\beta^\infty(\mathbf{x}(\{i, s_1'' \dots s_{M-1}''\}))(P_{s_0',i}^0 - \rho(i))$$

$$\le \max_{i \in \mathcal{S}} V_\beta^\infty(\mathbf{x}(\{i, s_1'' \dots s_{M-1}''\}))(1 - \sum_{i \in \mathcal{S}} \rho(i)) - \min_{i' \in \mathcal{S}} V_\beta^\infty(\mathbf{x}(\{i', s_1'' \dots s_{M-1}''\}))(1 - \sum_{i \in \mathcal{S}} \rho(i))$$

Note, by assumption (1), $\rho := \sum_{i \in \mathcal{S}} \rho(i) > 0$. Further, let

$$\sigma := \max_{\boldsymbol{x}, \boldsymbol{x}' \in \Delta_\mathcal{S}^{(M)}, \|\boldsymbol{x} - \boldsymbol{x}'\|_1 \le 2/M} |V_\beta^\infty(\boldsymbol{x}) - V_\beta^\infty(\boldsymbol{x}')|$$

Let us denote by $\tilde{\boldsymbol{x}} := \mathbf{x}(\{i, s_1'' \dots s_{M-1}''\})$ and $\tilde{\boldsymbol{x}}' := \mathbf{x}(\{i', s_1'' \dots s_{M-1}''\})$ we have: $\|\tilde{\boldsymbol{x}} - \tilde{\boldsymbol{x}}'\|_1 \le 2/M$. This leads us to the following result, for any $\boldsymbol{x}, \boldsymbol{x}'$ such that $\|\boldsymbol{x} - \boldsymbol{x}'\|_1 \le 2/M$ we have:

$$|V_\beta^\infty(\boldsymbol{x}) - V_\beta^\infty(\boldsymbol{x}')| \le \frac{1}{M} + (1 - \rho)\sigma$$

In particular, choose $\boldsymbol{x}, \boldsymbol{x}' \in \Delta_\mathcal{S}^{(M)}$ that maximize the value of $V_\beta^\infty(\boldsymbol{x}) - V_\beta^\infty(\boldsymbol{x}')$, we then have,

$$\sigma \le \frac{1}{M} + (1 - \rho)\sigma$$

Finally, putting the results together we have for any $\boldsymbol{x}, \boldsymbol{x}' \in \Delta_\mathcal{S}^{(M)}$ with $\|\boldsymbol{x} - \boldsymbol{x}'\|_1 \le 2/M$:

$$|V_\beta^\infty(\boldsymbol{x}) - V_\beta^\infty(\boldsymbol{x}')| \le \sigma \le \frac{1}{2\rho}\|\boldsymbol{x} - \boldsymbol{x}'\|_1$$

$$\square$$

We are now ready for the final steps of the proof.

*Proof of Lemma C.1.* The result then follows by noting that for any $\boldsymbol{x}, \boldsymbol{x}' \in \Delta_\mathcal{S}^{(M)}$ there exists a shortest path from $\boldsymbol{x}$ to $\boldsymbol{x}'$, say $\{\boldsymbol{x} = \boldsymbol{x}_0, \boldsymbol{x}_1, \boldsymbol{x}_2 \dots, \boldsymbol{x}_P = \boldsymbol{x}'\}$ such that between any two sequential components exactly one component is changed at a time $\|\boldsymbol{x}_i - \boldsymbol{x}_{i+1}\| = \frac{2}{M}$. Clearly, no more than $2M$ changes can occur along this path, it follows that $P < 2M$. Summing along this path allows us to see that for all $\boldsymbol{x}, \boldsymbol{x}' \in \Delta_\mathcal{S}^{(M)}$ we have,

$$|V_\beta^\infty(\boldsymbol{x}) - V_\beta^\infty(\boldsymbol{x}')| \le \frac{1}{\rho}\|\boldsymbol{x} - \boldsymbol{x}'\|_1$$

We can now complete the proof by noting that $\cup_{M>0}\Delta_\mathcal{S}^{(M)}$ is dense in $\Delta_\mathcal{S}$. Hence, the result must hold for any $\boldsymbol{x}, \boldsymbol{x}' \in \Delta_\mathcal{S}$. This completes the proof. $\square$

# F    DETAILS ABOUT THE EXPERIMENTS

## F.1    GENERALITIES

All our simulations are implemented in `Python`, by using `numpy` for the random generators and array manipulation and `pulp` to solve the linear programs. All simulations are run on a personal laptop (macbook pro from 2018). To ensure reproducibility, we will make the code and the Python notebook publicly available (this not done now for double-blind reasons).

**Value of $\tau$**  – Except specified otherwise, we use the value $\tau = 10$ in all of our example except for the example Yan (2022) where we use $\tau = 50$.

**Number of simulations and confidence intervals** – To obtain a estimate of the steady-state average performance, we simulate the system up to time $T = 1000$ and estimate the average performance by computing the average over values from $t = 200$ to $T = 1000$. All reported confidence intervals correspond to 95% interval on the mean computed by using $\hat{\mu} \pm 2\frac{\hat{\sigma}}{\sqrt{K-1}}$, where $\hat{\mu}$ and $\hat{\sigma}$ are the empirical mean and standard deviation and $k$ is the number of samples (the number of independent trajectories and/or number of randomly generated examples).

**A note on choice of algorithms for comparison** As specified in the main body of the paper, we choose to restrict our comarison to two algorithms: LP-update and FTVA. We note here that Hong et al. (2024a) is theoretically an exponentially optimal solution to the RMAB problem and its absence in our comparison might seem conspicuous to the reader. Hong et al. (2024a) is a *two set policy* which requires the *identification of a largest set* that can be aligned with the fixed point and actively steering the remaining arms towards this set. Theoretically, we know that for a suitable definition such a set always exists for any finite joint state, however, from a practical perspective the problem of choosing a "largest" set that aligns with the fixed point seems unclear to us. Unfortunately, the authors do not provide an implementation for their algorithm and hence, we refrain from comparing our work in order to avoid misleading our readers about the efficacy of their algorithm in comparison to our own as the performance of such an algorithm might heavily depend on the choice of hyperparameter used to implement the alignment procedure. On the contrary, the LP-update does not have any hyperparameters (except for $\tau$ that we set to $\tau = 10$ in all our simulations), and so have FTVA and LP-update.

## F.2 Example Hong et al. (2023)

This example has 8 dimensions and its parameters are:

$$
P^0 = \begin{pmatrix}
1 & & & & & & & \\
1 & & & & & & & \\
& 0.48 & 0.52 & & & & & \\
& & 0.47 & 0.53 & & & & \\
& & & & 0.9 & 0.1 & & \\
& & & & & 0.9 & 0.1 & \\
& & & & & & 0.9 & 0.1 \\
0.1 & & & & & & & 0.9
\end{pmatrix}
$$

$$
P^1 = \begin{pmatrix}
0.9 & 0.1 & & & & & & \\
& 0.9 & 0.1 & & & & & \\
& & 0.9 & 0.1 & & & & \\
& & & 0.9 & 0.1 & & & \\
& & & & 0.46 & 0.54 & & \\
& & & & & 0.45 & 0.55 & \\
& & & & & & 0.44 & 0.56 \\
& & & & & & & 0.43 & 0.57
\end{pmatrix}
$$

$R_i^1 = 0$ for all state and $R^0 = (0, 0, 0, 0, 0, 0, 0, 0.1)$.

For $\alpha = 0.5$ (which is the parameter used in Hong et al. (2023)), the value of the relaxed LP is $0.0125$ and the LP index are $[0.025, 0.025, 0.025, 0.025, 0, -0.113, -0.110, -0.108]$. Note that these numbers differ from the "Lagrangian optimal indices" given in Appendix G.2 of Hong et al. (2023). This is acknowledge in Hong et al. (2023) when the authors write "In [their] setting, because the optimal solution $y$ remains optimal even without the budget constraint, we can simply remove the budget constraint to get the Lagrangian relaxation [...] A nuance is that the optimal Lagrange multiplier for the budget constraint is not unique in this setting, so there can be different Lagrange relaxations". As a result, the priorities given by their indices and the true LP-index are different. In our simulation, we use the "true" LP-index and not their values. We also tested their values and obtained results that are qualitatively equivalent for the LP-priority (*i.e.*, for their order or ours, the LP-priority essentially gives no reward).

### F.3 EXAMPLE YAN (2022)

This example has $|\mathcal{S}| = 3$ dimensions and its parameters are:

$$P^0 = \begin{pmatrix} 0.022 & 0.102 & 0.875 \\ 0.034 & 0.172 & 0.794 \\ 0.523 & 0.455 & 0.022 \end{pmatrix} \qquad P^1 = \begin{pmatrix} 0.149 & 0.304 & 0.547 \\ 0.568 & 0.411 & 0.020 \\ 0.253 & 0.273 & 0.474 \end{pmatrix}$$

$R^1 = (0.374, 0.117, 0.079)$ and $R_i^0 = 0$ for all $i \in \mathcal{S}$.

For $\alpha = 0.4$ (which was the parameter used in Yan (2022)), the value of the relaxed LP is 0.1238 and its LP-index are $(0.199, -0.000, -0.133)$.

### F.4 RANDOM EXAMPLE

To generate random example, we use functions from the library `numpy` of `Python`:

- To generate the transition matrices, We generate an array of size $S \times 2 \times S$ by using the function `np.random.exponential(size=(S, 2, S))` from the library numpy of python and we then normalize each line so that the sum to $1$.
- We generate reward by using `np.random.exponential(size=(S, 2))`

The example used in Figure 1(a) corresponds to setting the seed of the random generator to 3 by using `np.random.seed(3)` before calling the random functions. Its parameters (rounded to 3 digits) are:

$$P^0 = \begin{pmatrix} 0.101 & 0.155 & 0.043 & 0.090 & 0.281 & 0.285 & 0.017 & 0.029 \\ 0.006 & 0.207 & 0.076 & 0.136 & 0.085 & 0.299 & 0.147 & 0.043 \\ 0.317 & 0.254 & 0.065 & 0.013 & 0.144 & 0.111 & 0.061 & 0.035 \\ 0.098 & 0.183 & 0.069 & 0.068 & 0.218 & 0.028 & 0.200 & 0.136 \\ 0.053 & 0.080 & 0.009 & 0.038 & 0.483 & 0.036 & 0.159 & 0.143 \\ 0.018 & 0.105 & 0.027 & 0.397 & 0.150 & 0.102 & 0.161 & 0.040 \\ 0.110 & 0.050 & 0.088 & 0.024 & 0.023 & 0.142 & 0.169 & 0.393 \\ 0.055 & 0.043 & 0.017 & 0.494 & 0.227 & 0.034 & 0.119 & 0.011 \end{pmatrix}$$

$R^0 = (0.073, 0.087, 0.778, 0.186, 1.178, 0.417, 1.996, 1.351)$

$$P^1 = \begin{pmatrix} 0.011 & 0.124 & 0.006 & 0.131 & 0.224 & 0.070 & 0.241 & 0.191 \\ 0.071 & 0.138 & 0.033 & 0.023 & 0.045 & 0.250 & 0.339 & 0.101 \\ 0.093 & 0.113 & 0.056 & 0.061 & 0.109 & 0.351 & 0.157 & 0.059 \\ 0.158 & 0.176 & 0.151 & 0.150 & 0.060 & 0.142 & 0.053 & 0.109 \\ 0.370 & 0.185 & 0.261 & 0.020 & 0.022 & 0.064 & 0.047 & 0.030 \\ 0.199 & 0.139 & 0.099 & 0.050 & 0.141 & 0.104 & 0.082 & 0.187 \\ 0.214 & 0.088 & 0.011 & 0.075 & 0.295 & 0.174 & 0.075 & 0.068 \\ 0.028 & 0.157 & 0.126 & 0.078 & 0.039 & 0.127 & 0.376 & 0.069 \end{pmatrix}$$

$R^1 = (0.059, 3.212, 1.817, 0.302, 2.259, 0.067, 0.344, 0.172)$.

For $\alpha = 0.5$, the value of the relaxed problem is $1.3885$ and the LP-index are $0.377, 3.273, 0.846, -0.116, 0.802, , -1.230, -0.562$.

For the other randomly generated examples, we compute the average performance over 50 examples by varying the seed between $0$ and $49$ for most of the figures except for Figures 4(b) where we only use 20 examples (and vary the seed between $0$ and $19$) to improve computation time.

