# OpenReview forum: "Model predictive control is almost optimal for restless bandits"
_ICLR.cc/2025/Conference — Submitted to ICLR 2025_

### Official Review · Reviewer_Q5VC · 2024-10-29

**Soundness:** 3
**Presentation:** 3
**Contribution:** 3
**Rating:** 6
**Confidence:** 3

**Summary:**

This paper proposes a new way to solve restless bandit: putting both the state information and the action information into a continuous vector space, and do linear programming to maximize the reward in the next $\tau$ rounds. When we look forward $\tau$ steps with a large enough $\tau$, this algorithm can find near optimal solutions. The authors show that normally the suboptimality gap is $O(\sqrt{1/N})$, but with certain conditions, it can be reduced to be $O(\exp(-cN))$, where $N$ is the number of arms and $c$ is some constant. As for the gap corresponding to $\tau$, they also show it is about $O({1\over \tau})$. In experiments, it is shown that choosing $\tau$ as a small value (e.g., 10) leads to very good performance, indicating the efficiency of model predictive control.

**Strengths:**

- The analysis shows a good suboptimality gap.

- The algorithm is quite efficient to implement.

- The experiment results are also convincing.

**Weaknesses:**

- The author assumes that all the arms are statistically identical. Is this a common assumption in restless bandits? I think there are many cases that the arms are not identical. It seems that your algorithm cannot be adapted to this setting easily (e.g., if all the arms have distinct transitions and rewards)?

**Questions:**

- Are there any comparison on the running time of different algorithms, and your algorithm with different $\tau$?

- Are there any experiment on real-data?

- In which part of the proof shows that $\tau(\epsilon) = O({1\over \epsilon})$?


=========After Rebuttal=========

Thanks for your reply, I do not have other questions.

---

> ### Author Response · Authors · 2024-11-18
>
> We would like to thank the reviewer for the encouraging comments and score.
>
> **Rev** *Are there any comparison on the running time of different algorithms, and your algorithm with different $\tau$?*
>
> In the submitted version of the paper, we do not compare the running time of our algorithm as a function of the parameters. One of the reasons for that our code uses a non-optimized Python implementation and we strongly believe that they are a lot of optimization that could be implemented to make our code run faster. Still, a quick benchmark seem to indicate that the computational complexity of our implementation grows around $\tau^2$: to compute one point of control, it takes a few milliseconds to solve the problem when $\tau=10$ and around $0.8$ seconds when $\tau=100$.
>
> **Rev**: *Are there any experiment on real-data?*
>
> **Anwer** As our paper is theoretical by nature, we do not made experiments on real-world data. Note that such experiments are rather uncommon in papers presenting theoretical results on restless bandit problems.
>
> **Rec**: *In which part of the proof shows that $\tau(\epsilon) = \mathcal{O}(1/\epsilon)$?*
>
> **Answer** Note that the asymptotic convergence result stems from $\sum_{t = 1}^{\infty} L_{t}(x) - L_{t - 1}(\Phi(x, u)) < \infty$ is bounded and $L_{t}(x) \geq L_{t - 1}(x)$. This means we have an infinite sum of non-negative numbers, hence, the $t^{th}$ term must "on average" fall faster than order $1/t$. A more technical way of writing the result may be to say that for a sequence of $\epsilon_i \to 0$, picking $\tau(\epsilon_i) = C/\epsilon_i$ results in $L_{\tau(\epsilon_i)}(x) - L_{\tau(\epsilon_i) - 1}(\Phi(x, u))$ being "frequently in" a ball of radius $\epsilon_i$ but such a statement would be needlessly difficult to read for an uninitiated reader.

---

### Official Review · Reviewer_QiUV · 2024-11-03

**Soundness:** 3
**Presentation:** 3
**Contribution:** 2
**Rating:** 5
**Confidence:** 4

**Summary:**

This paper addresses the infinite-horizon average reward RMAB problem. It proposed a model predictive control (MPC) policy using a rolling computational horizon of length $\tau$, which achieves a suboptimality gap with order $\mathcal{O}(1/\sqrt{N})$ with $N$ being the number of arms.

**Strengths:**

The paper presents a novel use of dissipativity for analyzing RMABs, which offers fresh insights into this field. The suboptimality bounds are rigorously derived, showcasing that the MPC-based policy approaches optimality as the number of arms, 𝑁, increases.

**Weaknesses:**

Though this paper conveys a solid algorithm design and theoretical proof, we must admit that the current submission has significant weaknesses.

1. My main concern is that the order of $\mathcal{O}(1/\sqrt{N})$ gap has been a well-known result for a long time, which various types of algorithms can achieve. In particular, the LP-based algorithms in many related works cited by this submission without an additional MPC layer can also achieve this optimality gap.  Though the reviewer admits that the proposed MPC layer can bring some advantages, the key reason that this reviewer does not champion this paper is that there is a new work (https://arxiv.org/pdf/2410.15003) recently has pushed the gap to the order of $\mathcal{O}(1/N)$ by using a diffusion approximation technique. I understand that the authors submitted their work earlier than this recent work and may not be aware of this new work, the main concern always holds as the proposed MPC-based algorithm does not improve the well-known optimality gap.

2.  Resolving the LP at each time step is not really a novel idea, which can be found in multiple works in related work cited by this submission. Though controlling $\tau$ is new, resolving LP at each time step causes a significant amount of computational complexity.  Aligned with my first concern, why do we need such a complex algorithm that does not even improve the theoretical guarantee, which is claimed as a main contribution in this work?

3. In many real-world applications that can be modeled as an RMAB framework, the underlying MDP for each arm may not be known in advance. My question is how can we leverage the proposed MPC-based algorithm for such a setting?

4. This paper considers a homogeneous arm setting. The reviewer quite doubts the scalability of the proposed algorithm when arms are heterogeneous and the number of arms is very large.

**Questions:**

Please refer to **Weaknesses**.

---

> ### Author Response · Authors · 2024-11-18
>
> Our main comments concern question (2) below for which we do not agree with the critic.
>
> **Rev** (1) My main concern is that the order of gap has been a well-known result for a long time, which various types of algorithms can achieve. In particular, the LP-based algorithms in many related works cited by this submission without an additional MPC layer can also achieve this optimality gap.
>
> **Answer** We agree that we do not improve on the $O(\sqrt{N})$ in the general case. Yet, we provide an algorithm that has many advantages:
> - It is remarkably simple and is very easy to implement without requiring difficult to tune parameters.
> - It has the same $O(\sqrt{N})$ guarantee (that is tight for our benchmark) in what seems the most general setting so far.
> - It performs extremly well in practice.
>
> **Rev** (2) *Though the reviewer admits that the proposed MPC layer can bring some advantages, the key reason that this reviewer does not champion this paper is that there is a new work (https://arxiv.org/pdf/2410.15003) recently has pushed the gap to the order $\mathcal{O}(1/N)$ of by using a diffusion approximation technique. I understand that the authors submitted their work earlier than this recent work and may not be aware of this new work, the main concern always holds as the proposed MPC-based algorithm does not improve the well-known optimality gap.*
>
> **Answer** We do not agree with this criticism, for multiple reasons:
> 1. First, as pointed out by the reviewer, the paper https://arxiv.org/pdf/2410.15003 appeared after our submission. Hence, this paper should not be considered as related work.
> 2. Second, even if it is, the problem studied in https://arxiv.org/pdf/2410.15003 is that of a **degenerate finite-horizon** problem. Our seting (average reward) concerns the infinite horizon problem. These two problems are not equivalent, in fact, in the absence of further assumptions (for e.g. UGAP) or our own proof techniques, the divergence of the finite horizon solution can be order exponential in the horizon (see for e.g. the value of the Lipschitz constant in Gast 2023 b). A natural follow up question might be, can Yan's method be appended to our own to achieve the same results? The answer to this is quite unclear: Yan et al's work requires a second $H$ horizon stochastic program solution, showing that such a solution satisfies the dissipativity property is highly non-trivial. Therefore, if reviewer's main concern is that an $\mathcal{O}(1/N)$ algorithm has already been discovered, they can rest assured that this does not hold under the infinite horizon average reward setting. We encourage the reviewer to reconsider their score.
>
> **Rev**: (3) *Resolving the LP at each time step is not really a novel idea, which can be found in multiple works in related work cited by this submission. Though controlling is new, resolving LP at each time step causes a significant amount of computational complexity. Aligned with my first concern, why do we need such a complex algorithm that does not even improve the theoretical guarantee, which is claimed as a main contribution in this work?*
>
> **Answer**. We think that the *beauty* of our solution is that ``it suffices to solve the natural LP at each time to obtain asymptotic optimality in the most general setting''.  We refer to our answer to point (1) and our general comment on the novelty of our result.
>
> **Rev** (4): *In many real-world applications that can be modeled as an RMAB framework, the underlying MDP for each arm may not be known in advance. My question is how can we leverage the proposed MPC-based algorithm for such a setting?*
>
> **Answer** We thank the reviewer for raising an interesting question. Our results can be used in model based reinforcement learning algorithms. For example: by using optimism in combination with our results on finite time horizon problems to approximate an equivalent cost function. It should be noted that we have shown the asymptotic convergence of the finite horizon rotated cost problem to the infinite horizon problem in our proof of Theorem 4.1 using dissipativity. This is an interesting line of future work which we intend to explore.
>
> **Rev** (5): *This paper considers a homogeneous arm setting. The reviewer quite doubts the scalability of the proposed algorithm when arms are heterogeneous and the number of arms is very large.*
>
> **Answer**: The reviewer raises another very interesting question. We would like to politely disagree with regards to the scalability issue in the heterogeneous case. We strongly suspect that our solutions can be used to come up with index based policies for the heterogeneous case but we do admit that we do not, as of this moment have a proof for this problem. This is a direction we are currently exploring.

---

> ### Author Response · Authors · 2024-11-25
>
> Dear Reviewer QiUV,
>
> We would like to thank you for your review, comments and excellent questions. We have tried to answer your questions as well as address your main concern regarding the rate of convergence of the optimal algorithm for the infinite horizon problem. As the time for open discussions are coming to an end, we would like to heavily encourage you to take a look at our response and let us know if there are any further questions that we may address. We would also like you to reconsider your score as we believe that your main concerns regarding our paper are unfounded. If there is any clarification that we could add, please ask.

---

> > ### Author Response · Authors · 2024-11-27
> > **Revisions based on your reviews**
> >
> > Dear Reviewer QiUV,
> >
> > We have highlighted our main contributions on page 2. We have also explicitly described the shortcomings of Yan et al's recent work in the average reward setting on page 13 of the appendix.
> >
> > Thank you

---

### Official Review · Reviewer_WFmj · 2024-11-03

**Soundness:** 2
**Presentation:** 1
**Contribution:** 2
**Rating:** 3
**Confidence:** 4

**Summary:**

This paper studied the discrete time infinite horizon average reward restless markovian bandit (RMAB) problem and focused on the asymptotic optimality in this setting. In particular, a model predictive control (MPC) based non-stationary policy with a rolling computational horizon $\tau$ is proposed and its sub-optimality gap is presented. The performance of this policy is also evaluated via simulations.

**Strengths:**

- RMAB has been extensively studied in recent years, from both the offline and online settings. This paper investigates the fundamental property of RMAB (e.g., the asymptotic optimality, optimality gap) in the offline setting. It is a challenging and interesting problem.
- A model predictive control (MPC) based non-stationary policy has been developed and theoretically analyzed, i.e., its optimality gap is characterized in terms of the number of arms $N$.
- Some experimental results were presented to validate the performance of this MPC based policy and its comparison with baselines.

**Weaknesses:**

- The asymptotic optimality performance of RMAB has raised many attentions in recent years. Despite that this paper proposed a policy by leveraging the MPC idea, it is hard to identify the technical novelty from the perspectives of algorithm design and technical proofs and results (See questions below). This paper heavily relies on the previous works such as Gast et al. 2023 a,b.
- The simulations are rather weak in both the settings and the baseline methods considered.
- This paper in general is poorly written with many typos, broken sentences and abused notations.

**Questions:**

- On one hand, the LP based relaxation has been extensively used in the RMAB literature, such as Verloop 2016, Zhang and Frazier 2022. On the other hand, the randomized rounding procedure is almost the same as that in Gast et al. 2023a and a finite-horizon MPC algorithm (LP-update policy) was proposed in Gast et al. 2023a,b. It is more like a straightforward extension. From the algorithmic perspective, can you more explicitly state what you see as the key novel aspects of the MPC based algorithm compared to prior work, particularly Gast et al. 2023a,b.?
- The first result in Section 4.1. This is not surprising and it is commonly known in the RMAB literature that LP-based method for RMAB is provably asymptotically optimal. Indeed, the LP based method has been leveraged to design index policy for RMAB problem without the hard-to-verify indexability condition as required by the Whittle index policy, and such a LP-based method to design index policy is provably asymptotically optimal as shown in the literature, e.g., Verloop 2016 is one of the first works in this domain.  Can you discuss how your result in Section 4.1 advances the state-of-the-art beyond what was already known from works like Verloop 2016? Are there any aspects of your analysis or bounds that are novel or improved?
- The second result in Section 4.2. Likewise, the proof is directly from Gast et al. 2023a and Hong et al. 2024a. For both results, can your clarify exactly how the use of dissipativity differs from or improves upon previous approaches? can you discuss any limitations of previous methods that your approach overcomes?
- In practice, how to determine how many arms to pull at each time, given that $\alpha N$ may not be an integer? If it is not an integer (may consider as an average constraint, there is no need to design an index policy).
-Equation (3) itself is not a RMAB problem. It should be properly defined with the budget constraint to be satisfied at teach time. It may be better to rigorously define (3)
- Many typos in the paper, just to name a few here: line 120, "the budget constraint, $\alpha$", line 146, $\boldsymbol{x}_s$ is not defined. line 144, since $u(s,a)$ is denoted as the empirical distribution of the state-action pairs $(s,1)$, why not just express it as $u(s,1)$ since the action is fixed.
- In Section 3.1 after introducing the optimization problem in (8), the authors claimed that this problem is computationally easy to solve. On one hand, indeed this is a linear problem and it is "relatively" easy to solve when the state space and the number of arms is small, given many LP solvers. On the other hand, this claim is not precise, since solving a LP with large-scale parameters/spaces can still be very computationally expensive and take a lot of time in practice. This may be one of the limitations of LP based method for RMAB compared to the Whittle index policy although LP based methods do not require the indexability condition.
- Can you elaborate why the definition of (8) imposes the constraint to be satisfied for each time $t$ as claimed in lines 198-199?

**Details Of Ethics Concerns:**

N/A.

---

> ### Author Response · Authors · 2024-11-18
> **Answer to comments (1/2)**
>
> We would like to emphasize that **some of the criticisms made are not correct**, and in particular the points 2 and 4 below.
>
> **Rev** (1): *On one hand, the LP based relaxation has been extensively used in the RMAB literature, such as Verloop 2016, Zhang and Frazier 2022. On the other hand, the randomized rounding procedure is almost the same as that in Gast et al. 2023a and a finite-horizon MPC algorithm (LP-update policy) was proposed in Gast et al. 2023a,b. It is more like a straightforward extension. From the algorithmic perspective, can you more explicitly state what you see as the key novel aspects of the MPC based algorithm compared to prior work, particularly Gast et al. 2023a,b.?*
>
> **Answer**: We thank the reviewer for the comment. The algorithm itself has been well established, the novelty lies in proving that a finite horizon LP-update policy returns an asymptotically optimal solution to an infinite horizon problem under minimal assumptions. Please take a look at the general comment for more details.
>
> **Rev** (2) *The first result in Section 4.1. This is not surprising and it is commonly known in the RMAB literature that LP-based method for RMAB is provably asymptotically optimal. Indeed, the LP based method has been leveraged to design index policy for RMAB problem without the hard-to-verify indexability condition as required by the Whittle index policy, and such a LP-based method to design index policy is provably asymptotically optimal as shown in the literature, e.g., Verloop 2016 is one of the first works in this domain. Can you discuss how your result in Section 4.1 advances the state-of-the-art beyond what was already known from works like Verloop 2016? Are there any aspects of your analysis or bounds that are novel or improved?*
>
> **Answer**: We do not agree with this criticism by the reviewer. Yes, Verloop 2016 proposes a solution that does not require indexability, but this paper requires the  condition ``UGAP'' that is known to be very difficult to verify. Moreover, Verloop 2016 does not provide a rate of convergence. See our "general comments" for more details.
>
> **Rev** (3): *The second result in Section 4.2. Likewise, the proof is directly from Gast et al. 2023a and Hong et al. 2024a. For both results, can your clarify exactly how the use of dissipativity differs from or improves upon previous approaches? can you discuss any limitations of previous methods that your approach overcomes?*
>
> **Answer**: With regards to Section 4.2 we note that due to the results from section 4.1 and continuity of the rotated cost function, we can conclude that the LP-update policy does *steer* the state towards the fixed point *only by assuming the uniqueness of the fixed point*. Hence, while parts of the proof are adapted from Gast et al. 2023a and Hong et al. 2024a, the two main steps that allow us to conclude that the state will lie in a local ball around the fixed point after a finite time are entirely an outcome of our dissipativity idea. This should be compared to the proofs in Gast et al. 2023a which *assumed UGAP* or Hong et al. 2024a which *assumed local stability of the policy* and aperiodicity of the transition kernel *induced by the policy* to make this claim.
>
> **Rev** (4): *In practice, how to determine how many arms to pull at each time, given that may not be an integer? If it is not an integer (may consider as an average constraint, there is no need to design an index policy). -Equation (3) itself is not a RMAB problem. It should be properly defined with the budget constraint to be satisfied at each time. It may be better to rigorously define (3)*
>
> **Answer**. We think that there is a missunderstanding: When looking at the randomized rounding policy (in our appendix), it clearly uses the floor function whenever $\alpha N$ is not an integer and this is reflected in the bound for Theorem 4.1.  The reviewer should note that in the paragraph above equation (2) we clearly restrict our policy space to policies that are stationary and satisfy the budget constraint of pulling at most $\alpha N$ arms. This definition of the policy space $\Pi^{(N)}$ characterizes the Restless bandit problem.
>
> **Rev** (5): *There are many typos in the paper.*
>
> **Answer**: We thank the reviewer for pointing out the typos and will make an effort to correct them.

---

> > ### Author Response · Authors · 2024-11-18
> > **Answre to comments (2/2)**
> >
> > **Rev** (6): *In Section 3.1 after introducing the optimization problem in (8), the authors claimed that this problem is computationally easy to solve. On one hand, indeed this is a linear problem and it is "relatively" easy to solve when the state space and the number of arms is small, given many LP solvers. On the other hand, this claim is not precise, since solving a LP with large-scale parameters/spaces can still be very computationally expensive and take a lot of time in practice. This may be one of the limitations of LP based method for RMAB compared to the Whittle index policy although LP based methods do not require the indexability condition*.
> >
> > **Answer**: We would politely like to disagree with the reviewers claim regarding index policies. Even though a sub-cubic algorithms allows us to resolve indexability and compute the index, it is insufficient to prove the optimal nature of the Whittle's index (or LP-index) policy. As early as 1989, Weber and Weiss showed a counter example where an indexable system gave sub-optimal value under Whittle's index policies. It is critical to note the role of UGAP like additional assumptions in resolving this gap in performance. We do agree with the reviewer that when the appropriate assumptions hold, an indexing policy needs to compute the index once at the beginning and no new computations need to be made except arranging the arms according to the priority order.
> >
> >
> > As pointed out in the answer to reviewer kuLz, we did some emperical test on the time to solve an LP problem by using the default LP-solver of PuLP (a Python library). Our implementation (that is not specifically optimized) takes roughly $(|S| \tau)^2$ micro-seconds computation to compute a policy at each step. There seems to be a natural trade-off between the strength of assumptions made on the system and the complexity of the algorithm required to achieve optimality both conceptually and computationally.
> >
> >
> > **Rev** (7): *Can you elaborate why the definition of (8) imposes the constraint to be satisfied for each time as claimed in lines 198-199?*
> >
> > **Answer**: Equation (8b) constrains the actions so that equation (4) holds at each time step i.e, $u \in \mathcal{U}(x)$ which means that at most $\alpha$ fraction of arms at each time can be pulled and if we have $x_s$ fraction of arms in state $s$, then no more than $Nx_s$ arms can be pulled from state $s$. This restriction is true for *each time step*.

---

> ### Author Response · Authors · 2024-11-25
>
> Dear Reviewer WFmj,
>
> We would like to thank you for your thorough review of our paper. We encourage you to take a careful look at our responses since it seems there are a number of misunderstandings regarding our technical contributions as well as its place in the current literature of restless bandits. In particular we encourage you to take a look at our response to point 2 and 4 in our rebuttal. As the time for open discussions are coming to an end, we hope that you can respond to our replies or provide more points that might need to be addressed in our paper. We are hoping that our thorough response may have cleared some of your concerns and would like to encourage you to please reconsider your score. If there is any clarification that we could add, please ask.

---

> > ### Comment · Reviewer_WFmj · 2024-11-26
> > **Thank you**
> >
> > Thank you for the rebuttal which addressed some of my questions. I respectfully disagreed with the authors on the computationally efficient of the algorithms. No matter using the Python solver PuLP or the solvers from Gurobi, when the state space is large, it is known that it take "long" time to solve the problem. For many RMAB applications e.g., cloud computing, resource allocation, healthcare, the state space is often large in practice. The reviewer acknowledged some technical contributions in the paper, however, strongly doubted the importance of such results or the benefits to the community of using RMAB framework to solving real-world problems due to the computational complexity of the solutions.
> >
> > Note that ICLR allows paper revisions. However, the reviewer did not see any effort by the authors to improving the paper given that there are many typos and many part are poorly written. How could we believe the statement "will make an effort to correct them."
> >
> > In addition, I fully agree that the paper https://arxiv.org/pdf/2410.15003 appeared after the ICLR deadline, and this paper should not be criticized for not citing it. Once again, ICLR allow revisions, and the revisions should discuss related work properly.

---

> > > ### Author Response · Authors · 2024-11-26
> > > **Thank you for engaging in the discussion**
> > >
> > > About the computational efficiency: we agree with you that the solution that we propose, and that consists in resolving a problem at each time, will be slower than an index-based method. Our implementation suggest that this takes (T\tau)^2 micro-seconds at every decision epoch, which means that one can hardly imagine to use this solution for a problem of dimension more than 100 or 1000.  Note that for Whittle index, it is possible to compute the index of models of up to 1000 or 10000 states but not really more than that unless a close form exist. The advantage of index policies is that the index have to be computed only once.
> > >
> > > That being said, the main contribution of our paper is theoretical and to show that the very natural LP-update framework can be shown to be asymptotically optimal. We do so by using what we think is an interesting framework for the community.  From a practical point of view, we are not claiming that everyone should use an LP-update like policies for bandits. Our message is rather that, for difficult problems where index fail, an LP-update approach can be valuable as it gives a much better performance than everything else.
> > >
> > > About the revision: we are currently working on an updated version but we did not have the time to converge yet. We wanted to avoid uploading too many versions and only update one that we think is ready. We will upload it soon.

---

> > > > ### Author Response · Authors · 2024-11-27
> > > > **Revisions based on your review**
> > > >
> > > > Dear Reviewer WFmj,
> > > >
> > > > Apart from the typos and sentence sculpting based on your recommendations we have also added a few lines of clarification on page 3 and page 6. Please let us know if others are found. More importantly, as mentioned in the general comments we added a paragraph highlighting our own contributions and its place in the literature on page 2. Further, we explicitly described the shortcomings of the recent work on diffusion and the $\mathcal{O}(1/N)$ error rate by Yan et al for our own work in the appendix on page 13 of the appendix.
> > > >
> > > > Thank you

---

### Official Review · Reviewer_kuLz · 2024-11-04

**Soundness:** 4
**Presentation:** 3
**Contribution:** 3
**Rating:** 6
**Confidence:** 3

**Summary:**

The paper addresses the discrete time infinite horizon average reward Restless Markovian Bandit (RMAB) problem with a Model Predictive Control (MPC) approach. The proposed MPC algorithm achieve the suboptimality gap $O(1/\sqrt{N})$ with a minimal set of assumptions, and can achieve exponential convergence under local stability conditions. Moreover, the MPC algorithm works well in practice with SOTA computational complexity.

**Strengths:**

1.	The paper introduces a novel application of MPC to the RMAB problem which achieve suboptimal gap $O(1/\sqrt{N})$ with a minimal set of assumptions and exponential convergence rate under local stability conditions.
2.	The proposed MPC approach reduces the computational burden associated with solving RMAB problems and perform well in numerical experiments.
3.	This paper presents an interesting framework based on dissipativity, and provides theoretical analysis in this paper.

**Weaknesses:**

1.	In Section 6, the algorithm LP-priority is not formally introduced. It is confusing to distinguish between the LP-update and LP-priority algorithms due to the lack of a clear definition.
2.	What are the main technical contributions of the theoretical analysis compared to existing works? I suggest highlighting the novelty and primary contributions of the theoretical analysis more clearly.
3.	In Line 88, the paper states, "It performs well both in terms of the number of arms N as well as the computational time horizon T, beating state-of-the-art algorithms in our benchmarks." However, in the numerical experiments section, the authors did not compare the computational efficiency of the proposed MPC approach with existing algorithms. Moreover, it would be beneficial to provide a more rigorous discussion on why the MPC approach reduces the computational burden.

**Questions:**

1.	In Section 6, the algorithm LP-priority is not formally introduced. It is confusing to distinguish between the LP-update and LP-priority algorithms due to the lack of a clear definition.
2.	What are the main technical contributions of the theoretical analysis compared to existing works? I suggest highlighting the novelty and primary contributions of the theoretical analysis more clearly.
3.	In Line 88, the paper states, "It performs well both in terms of the number of arms N as well as the computational time horizon T, beating state-of-the-art algorithms in our benchmarks." However, in the numerical experiments section, the authors did not compare the computational efficiency of the proposed MPC approach with existing algorithms. Moreover, it would be beneficial to provide a more rigorous discussion on why the MPC approach reduces the computational burden.

---

> ### Author Response · Authors · 2024-11-18
>
> **Rev**: *In Section 6, the algorithm LP-priority is not formally introduced. It is confusing to distinguish between the LP-update and LP-priority algorithms due to the lack of a clear definition.*
>
> **Answer**: We thank the reviewer for the comment, we will make changes to our simulation results to address their comment.
>
> **Rev**: *What are the main technical contributions of the theoretical analysis compared to existing works? I suggest highlighting the novelty and primary contributions of the theoretical analysis more clearly.*
>
> **Answer**: Please refer to the general comment regarding our contributions. One of our the main technical novelties is the use of the dissipativity framework to analyze model predictive control in the restless bandit context.
>
> **Rev**: *In Line 88, the paper states, ``It performs well both in terms of the number of arms N as well as the computational time horizon T, beating state-of-the-art algorithms in our benchmarks." However, in the numerical experiments section, the authors did not compare the computational efficiency of the proposed MPC approach with existing algorithms. Moreover, it would be beneficial to provide a more rigorous discussion on why the MPC approach reduces the computational burden.*
>
> **Answer:** We agree that we do not discuss explicitly the computational complexity of our solution. The main reason for not providing measure of the time complexity is that we use a simple python implementation that we did not try to optimize.  Yet, to give an order of magnitude, in all of the examples studied, the time to compute one control is around a few tens of milliseconds.

---

> > ### Author Response · Authors · 2024-11-25
> >
> > Dear Reviewer kuLz,
> > We thank you for your review. We hope that our response to your concerns have been satisfactory. As the time for open discussions are coming to a close, in case there are more clarifications to be added please let us know. We hope that we can address any further clarifications that need to be made.

---

> > > ### Author Response · Authors · 2024-11-27
> > > **Specific revisions based on your review**
> > >
> > > Dear Reviewer kuLz,
> > >
> > > In section 6, page 9, we added a paragraph explaining both the LP-priority as well as the FTVA algorithm which we used to compare our work. As mentioned in the general comments we highlighted the main contributions on page 2.
> > >
> > > Thank you

---

### Author Response · Authors · 2024-11-18
**General answer**

We thank the reviewers for their consideration and time.  We found these reviews constructive and quite postive (despite what we think is a misunderstanding with Rev WFmj). Before answering the specific comments of each reviewers after each review, we first make a general comment that answers the concern of multiple reviewers about the novelty of our results.

Broadly speaking, the literature on heuristics for time-average restless bandits can be decomposed in two kinds of algorithms:
- Algorithms that perform well in practice but require UGAP (e.g., Whittle index, the algorithm from Verloop 16)
- Algorithms that do not require UGAP but that do not perform well in practice (Hong et al. 23, 24, Avrachenkov 24, Yan 24)

The main contribution of our paper is the proof that a very natural model predictive control (LP-update) provides a best-of-both-worlds solution with minimal assumptions. We are not claiming that this algorithm is new, as the idea of resolving an LP for finite-horizon restless bandit already exists in the literature. Yet, all the papers that proposed to use this idea analyzed the algorithm in the finite-horizon case. The main reason for this is that without the framework of dissipativity, the analysis of the time-average case is hard. Note, in general a finite horizon policy may not even be optimally operated at the fixed point. The use of this framework is one of the key technical novelties of our approach.


**Further comparison with related work**: Some reviewers have noted the works of Verloop 2016, Zhang and Frazier 2022 as examples for the LP-based policies that would work in our setting. This is not true unless an additional condition known as Uniform Global Attractor Property (UGAP) is satisfied. The UGAP assumption is extremely difficult to verify and weakening such an assumption has been the focus of much of the recent literature in Restless bandits. Motivated by the idea that the fixed point is the optimal operating point for restless bandit problem, the first paper to break the UGAP requirement was Hong et al (2023) that used the so-called synchronization assumption. Yan 2024 used a reachability condition while Hong et al (2024) used aperiodicity and local attractor conditions, Avrachenkov 2024 used a fluid policy convergence assumption. Please take a look at the related work section for a more careful look at recent literature on these assumptions and their motivation. To the best of our knowledge all these assumptions are on both the system parameters *as well as on the policy used*, more importantly, these assumptions are stronger than the assumptions we use on the parameters.

On the other hand, Section 4.1. shows that a very weak assumption on the system parameters (the weakest to the best of our knowledge) is sufficient to ensure the optimality of the LP-update policy. Our work not only has the weakest assumptions on the system parameter (the assumption on the ergodicity coefficient Assumption 1.) but also makes no assumptions on the policy space, such assumptions are easily verifiable when the system parameters are known. In order to bridge this gap from finite horizon to infinite horizon problems we use dissipativity to show that the equivalent rotated cost minimization problem is monotone increasing but bounded. This critical insight allows us to use the value function instead of proving that our algorithm *steers* the state space towards the fixed point or needing to show that the policies from a finite horizon setting align with the infinite horizon optimal policy.

In conclusion, our result on the asymptotic optimality of the LP-update is actually very surprising since it suggests that a weak ergodicity assumption on the single arm problem without additional assumptions on the policy is sufficient to ensure convergence of the value function. From a more technical perspective, returning to the value function instead of looking at controllability in policy spaces can be very advantageous since the $\arg \max$ function used to find policies typically does not have good continuity properties but the value function retains such properties very well.

We will add these remarks in our main article to make these statements clear to our readers.

---

> ### Author Response · Authors · 2024-11-27
> **Regarding the Revision**
>
> Dear Reviewers,
>
> In light of your reviews we have made a few changes to our draft. Firstly, we have gone over and tried to correct typos where we could find them as well as modified some sentences slightly to reduce space consumption or slightly improve the phrasing. Such changes are not highlighted in a different color. Secondly, we have inserted a few major changes, these changes are highlighted in blue in our new draft. Of particular importance is the change made on page 2 which highlights the place of our work in the current literature. This is in accordance to the general comment we made to all the reviewers, the other comparisons can be found in the additional related works section in Appendix A.
>
> Thank you.

---

### Meta-Review · Area_Chair_Gnm9 · 2024-12-21

**Metareview:**

This is a borderline paper. Overall, the reviewers are critical of various aspects of the paper. Most notably about the somewhat incremental novelty and computational scalability issues. I therefore believe that a major revision of the paper is necessary. The reviews contain a variety of suggests of how to improve and revise the paper.

**Additional Comments On Reviewer Discussion:**

There was a significant discussion among reviewers and between reviewers and authors.

---

### Decision · Program_Chairs · 2025-01-22

Reject